# Mechanisms underlying microglial colonization of developing neural retina in zebrafish

**Nishtha Ranawat[†], Ichiro Masai***

Developmental Neurobiology Unit, Okinawa Institute of Science and Technology Graduate University, Onna, Japan

**Abstract** Microglia are brain-resident macrophages that function as the first line of defense in brain. Embryonic microglial precursors originate in peripheral mesoderm and migrate into the brain during development. However, the mechanism by which they colonize the brain is incompletely understood. The retina is one of the first brain regions to accommodate microglia. In zebrafish, embryonic microglial precursors use intraocular hyaloid blood vessels as a pathway to migrate into the optic cup via the choroid fissure. Once retinal progenitor cells exit the cell cycle, microglial precursors associated with hyaloid blood vessels start to infiltrate the retina preferentially through neurogenic regions, suggesting that colonization of retinal tissue depends upon the neurogenic state. Along with blood vessels and retinal neurogenesis, IL34 also participates in microglial precursor colonization of the retina. Altogether, CSF receptor signaling, blood vessels, and neuronal differentiation function as cues to create an essential path for microglial migration into developing retina.

**\*For correspondence:**
masai@oist.jp

**Present address:** [†]Neural Connectivity Development in Physiology and Disease Laboratory, Burke Neurological Institute, White Plains, United States

**Competing interest:** The authors declare that no competing interests exist.

## Editor's evaluation

The authors have addressed the remaining concerns raised by reviewers and the revised manuscript has been strengthened with revisions to the text and figures.

This manuscript will be of use to developmental neurobiologists and provides new insight on the mechanisms and microglia-vascular interactions and microglial colonization of the zebrafish retina.

## Introduction

Microglia are the resident macrophages of brain. These dedicated CNS phagocytes form the innate immune system of embryonic and adult brain. Microglia eliminate cellular debris to prevent neuro-inflammation and to promote neuronal protection in vertebrates (*Ashwell, 1991*; *Calderó et al., 2009*; *Lawson et al., 1990*; *Neumann et al., 2009*; *Sierra et al., 2010*). They also prune unnecessary synapses to establish functional, mature neural circuits during brain development, performing a variety of cellular functions (*Paolicelli et al., 2011*; *Tremblay et al., 2010*). In contrast to other CNS cells, like neurons and astrocytes, microglia do not originate from neural plate, but are derived from mesoderm (*Ashwell, 1991*; *Boya et al., 1979*) through hematopoiesis (*Ginhoux et al., 2013*). In developing zebrafish, embryonic hematopoiesis occurs in successive waves that are separated anatomically and temporally. The primitive or first wave of microglial precursors is generated from myeloid cells originating in the rostral blood island (RBI) at about 11 hr post-fertilization (hpf) (*Stachura and Traver, 2011*; *Xu et al., 2012*). The definitive wave is contributed by the ventral wall of the dorsal aorta (VDA), giving rise to hematopoietic stem cells (HSCs) (*Xu et al., 2015*). In addition, a short intermediate wave also originates from the posterior blood island (PBI) (*Bertrand et al., 2007*). After 2 weeks

**eLife digest** The immune system is comprised of many different cells which protect our bodies from infection and other illnesses. The brain contains its own population of immune cells called microglia. Unlike neurons, these cells form outside the brain during development. They then travel to the brain and colonize specific regions like the retina, the light-sensing part of the eye in vertebrates.

It is poorly understood how newly formed microglia migrate to the retina and whether their entry depends on the developmental state of nerve cells (also known as neurons) in this region. To help answer these questions, Ranawat and Masai attached fluorescent labels that can be seen under a microscope to microglia in the embryos of zebrafish. Developing zebrafish are transparent, making it easy to trace the fluorescent microglia as they travel to the retina and insert themselves among its neurons.

Ranawat and Masai found that blood vessels around the retina act as a pathway that microglia move along. Once they reach the retina, the microglia remain attached and only enter the retina at sites where brain cells are starting to mature in to adult neurons. Further experiments showed that microglia fail to infiltrate and colonize the retina when blood vessels are damaged or neuron maturation is blocked.

These findings reveal some of the key elements that guide microglia to the retina during development. However, further work is needed to establish the molecular and biochemical processes that allow microglia to attach to blood vessels and detect when cells in the retina are starting to mature.

post-fertilization, VDA-derived microglia progressively replace RBI-derived microglia throughout the CNS (*Ferrero et al., 2018*; *Xu et al., 2015*). Thus, primitive and definitive hematopoiesis contribute embryonic and adult microglia, respectively, during zebrafish development.

Generation of embryonic microglial precursors and their colonization of brain areas has been extensively described in zebrafish (*Herbomel et al., 2001*). In zebrafish, embryonic microglial precursors are initially specified in lateral plate mesoderm and then spread on yolk. They start to migrate into the cephalic mesenchymal region after 22 hpf. At 26–30 hpf, a few microglia are observed in the vitreous space or choroid fissure of the optic cup, and around 30 microglia colonize the neural retina by 48 hpf. Microglial colonization of the optic tectum and other regions of zebrafish brain occurs after 48 hpf, indicating that the retina is one of the first brain regions to be colonized by microglia during development.

Previous studies have suggested various signals that promote microglial colonization in brain. In mice, Cxcl12/CxcR4 signaling orchestrates microglial migration into developing cerebral cortex (*Arnò et al., 2014*; *Hattori and Miyata, 2018*). In zebrafish, microglia migrate from the yolk-sac and colonize the brain in an apoptosis-dependent manner (*Casano et al., 2016*; *Xu et al., 2016*). Microglial precursors also migrate into the cephalic mesenchymal area in a Colony Stimulating Factor-Receptor (CSF-R)-dependent manner (*Herbomel et al., 2001*; *Wu et al., 2018*). Zebrafish *fms* mutants carry a genetic mutation in CSF-R and show severe delays in microglial colonization of both brain and retina, as well as an increase in neuronal apoptosis (*Herbomel et al., 2001*). Recently, it was reported that brain colonization by microglial precursors depends primarily on one zebrafish CSF-R, CSF1ra, and one CSF-R ligand, IL34, and that this combination of CSF ligand and receptor dominates this process (*Wu et al., 2018*). Importantly, the number of microglia in the brain and retina is reduced in zebrafish *il34* mutants that overexpress anti-apoptotic protein, Bcl2. Thus, apoptosis and the IL34-CSF1ra signaling pathway cooperate to promote microglial colonization of the brain and retina during zebrafish development.

In developing zebrafish retina, neurogenesis is initiated in the ventro-nasal retina, adjacent to the optic stalk at 25 hpf and progresses to the whole region of the neural retina, suggesting a spatio-temporal pattern of retinal neurogenesis in zebrafish (*Hu and Easter, 1999*; *Masai et al., 2000*). Retinal progenitor cells are multipotent and give rise to six major classes of neurons and one type of glial cells. Two types of photoreceptors, rods, and cones, form the outer nuclear layer (ONL). Three interneurons, amacrine cells, bipolar cells, and horizontal cells form the inner nuclear layer (INL). Retinal ganglion cells (RGCs) form the RGC layer. Synaptic connections between photoreceptors and bipolar/horizontal cells form the outer plexiform layer (OPL), and synaptic connections between

RGCs and bipolar/amacrine cells form the inner plexiform layer (IPL). Cell fate determination is less dependent on the cell lineage of retinal progenitor cells, suggesting that both extrinsic and intrinsic mechanisms influence the status of retinal progenitor multipotency, leading to generation of diverse retinal cell types (*He et al., 2012*). These developmental profiles of retinal neurogenesis and cell differentiation may be coupled with microglial colonization. Although apoptosis and CSF-R signaling are suggested in microglial colonization of the retina in zebrafish (*Wu et al., 2018*), the mechanism underlying microglial colonization of the retina remains to be determined.

In this study, using zebrafish, we examined the developmental profile of retinal colonization by microglia precursors. The number of ocular microglial precursors progressively increases from 32 to 54 hpf. Most microglial precursors do not proliferate, suggesting that microglial colonization of the retina depends on cell migration from outside the optic cup. We found three guidance mechanisms driving microglial precursor colonization of the retina. First, IL34 initiates microglial precursor movement from yolk toward the brain and the retina. Second, microglia precursors enter the optic cup via ocular hyaloid blood vessels in the choroid fissure, suggesting that these blood vessels guide microglia to the retina. Third, microglial precursors infiltrate the neural retina preferentially through the neurogenic region, suggesting that the neurogenic state of retinal tissue acts as an entry signal for microglial precursors to infiltrate the retina. Thus, a series of guidance mechanisms promote microglial colonization from yolk to the neural retina in zebrafish.

## Results

### Embryonic microglial precursors progressively colonize developing zebrafish retina

In zebrafish, early macrophages are generated from myeloid cells originating in the RBI around 11 hpf and these macrophages colonize the brain and retina by 55 hpf (*Xu et al., 2015*). Around 60 hpf, brain and retina-resident macrophages undergo a phenotypic transition, which indicates expression of mature microglial markers, such as apolipoprotein E (apo E) and phagocytic behavior toward dead cells (*Herbomel et al., 2001*). Importantly, early macrophages outside the brain never express apo E (*Herbomel et al., 2001*), suggesting that only brain and retina-resident macrophages give rise to microglia. Thus, early macrophages localized in the brain and retina by 60 hpf are generally accepted as microglial precursors in zebrafish. In this study, we focused on two macrophage markers, *macrophage expressing gene 1.1* (*mpeg1.1*) (*Ellett et al., 2011*) and *microfibrillar-associated protein 4* (*mfap4*) (*Walton et al., 2015*), and define *mpeg1.1; mfap4*-positive cells inside the optic cup as microglial precursors colonizing the zebrafish retina.

To ascertain how microglia precursors migrate from peripheral tissues into the neural retina during development, we generated a zebrafish transgenic line, *Tg[mpeg1.1:EGFP],* using the original DNA construct (*Ellett et al., 2011*). As previously reported (*Ellett et al., 2011*), our established transgenic line visualized ocular microglial precursors and enabled us to monitor their number and location in the optic cup from 24 to 54 hpf. Accordingly, we obtained 3D images using confocal laser scanning microscopy (LSM) (*Figure 1A and B*). The first microglial precursor cells appeared near the choroid fissure and lens around 30–32 hpf. After that, the number of ocular microglial precursors increased to 19.1 ± 1.26 at 42 hpf and 31.0 ± 4.44 at 54 hpf (*Figure 1C*), indicating a progressive increase in the number of ocular microglial precursors. We also confirmed a similar progressive increase in the number of ocular *L-plastin*-positive cells, although *L-plastin* is expressed in microglial precursors and neutrophils in zebrafish brain (*Figure 1—figure supplement 1*). Next, to determine more precisely the spatial distribution of microglial precursors in the optic cup, we generated another transgenic line, *Tg[mfap4:tdTomato-CAAX],* using the original DNA construct (*Walton et al., 2015*). As previously reported (*Walton et al., 2015*), our established transgenic line efficiently labeled ocular microglial precursor membranes. We labeled this transgenic embryo using Bodipy ceramide conjugated with fluorescent Alexa-488, which visualizes retinal layer structures (*Figure 1—figure supplement 2*). From 32 to 36 hpf, mfap4$^+$ cells were mostly located in the vitreous space between the neural retina and lens, and possibly associated with ocular blood vessels, which develop around the lens. In 42–44-hpf retina, a few microglial precursor cells start to enter the neural retina and spread toward the emerging IPL, where they are associated with newly born amacrine cells (*Figure 1—figure supplement 3*). By 54 hpf, IPL formation is complete and microglial precursors were observed throughout all retinal tissue,

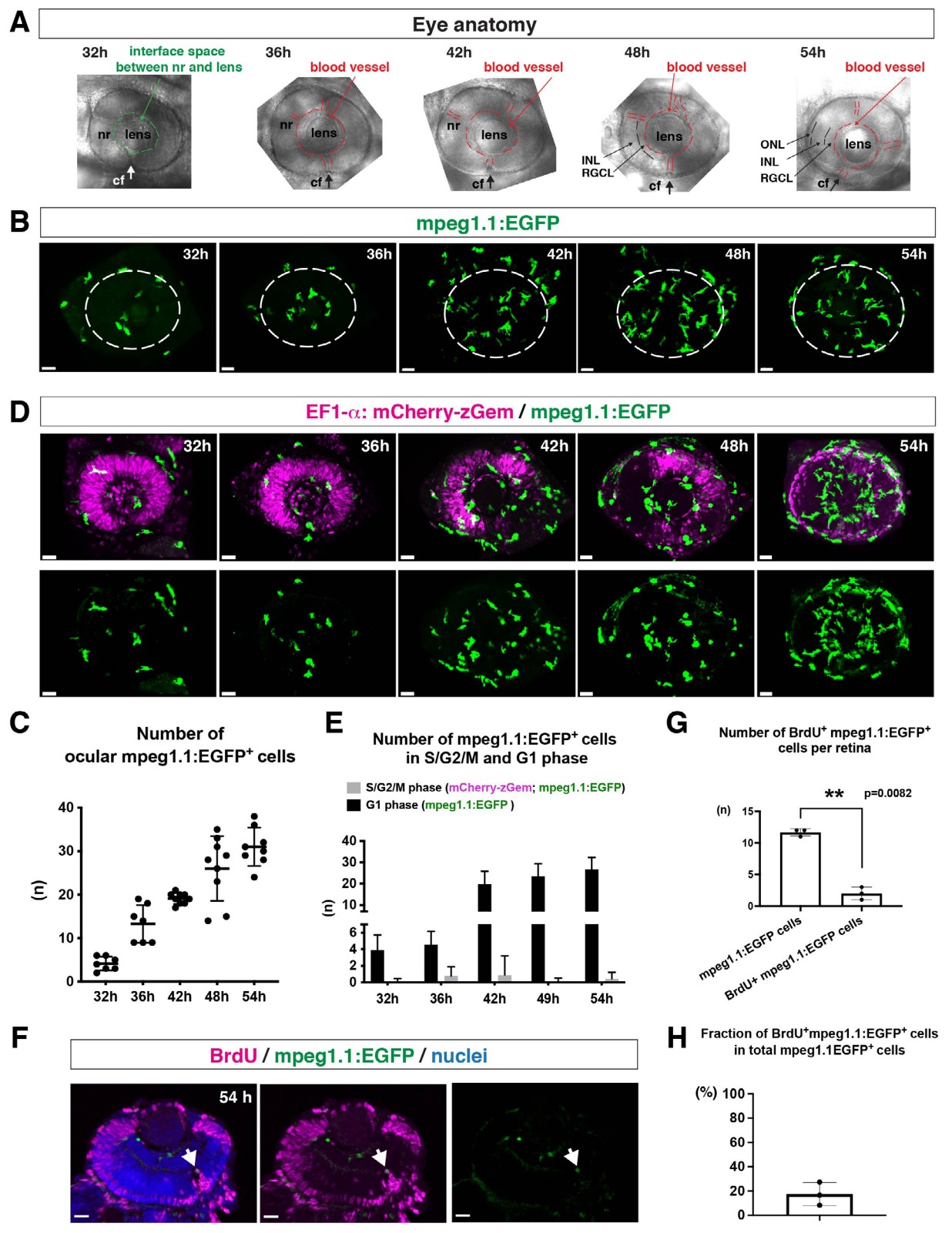

**Figure 1.** Microglial precursors progressively colonize developing zebrafish retinas. (**A**) Lateral view of zebrafish eyes used for confocal scanning shown in panel (**B**). Anterior is left and dorsal is up. The choroid fissure (cf, arrows) is formed at the ventral retina. At 32 hpf, the interface space between the neural retina (nr) and lens appears, in which ocular blood vessels are formed after 36 hpf. At 48 hpf, RGCL and INL are distinct. At 54 hpf, the ONL becomes evident. (**B**) Three-dimensional confocal images of mpeg1.1:EGFP-positive microglial precursors (green) in the retina from 32 to 54 hpf. Dotted

*Figure 1 continued on next page*

*Figure 1 continued*

circles indicate the outline of the optic cup. The first microglial precursors appear in the choroid fissure and near the lens at 32 hpf. Microglial precursors in the optic cup progressively increase in number. At 42 hpf, they start to enter retinal tissue and spread into the entire neural retina by 54 hpf. Scale: 30 μm. (**C**) Histogram of the number of intraocular microglial precursors from 32 to 54 hpf. Horizontal and vertical bars indicate means ± SD. (**D**) Three-dimensional confocal images of *Tg[EF1α:mCherry-zGem; mpeg1.1:EGFP]* retinas from 32 to 54 hpf. *Tg[EF1α:mCherry-zGem]* (magenta) indicates cells undergoing S and G2 phases. mpeg1.1:EGFP-positive microglial precursors (green) are mostly negative for mCherry-zGem, suggesting that most ocular microglial precursors are in G1 phase. Scale: 30μm. (**E**) Histogram of numbers of intraocular microglial precursors expressing only mpeg1.1:EGFP, and microglial precursors expressing both mCherry-zGem and mpeg1.1:EGFP in retinas from 32 to 54 hpf. Double-positive microglial precursors represent proliferating microglial precursors undergoing S/G2 phase. Single mpeg1.1:EGFP-positive microglial precursors represent microglial precursors in G1 phase. Bars and lines indicate means ± SD. (**F**) Sections of *Tg[mpeg1.1:EGFP]* transgenic retinas with BrdU incorporated and labeled with anti-BrdU (magenta) and anti-EGFP (green) antibody. Nuclei were counterstained by TOPRO3 (blue). The arrowhead indicates BrdU- and mpeg1.1:EGFP double-positive cells. Most mpeg1.1:EGFP$^+$ cells are BrdU-negative at 54 hpf, suggesting that they are not proliferative. Scale: 20 μm. (**G**) Histogram of numbers of mpeg1.1:EGFP$^+$ cells and BrdU-positive mpeg1.1:EGFP$^+$ cells per retinal section. Bars and lines indicate means ± SD. **p < 0.01. (**H**) Fraction of BrdU-positive proliferative mpeg1.1:EGFP$^+$ cells in total mpeg1.1:EGFP$^+$ cells. The average is less than 20%, indicating that more than 80 % of ocular microglial precursors are in the G1 phase. Bars and lines indicate means ± SD.

The online version of this article includes the following figure supplement(s) for figure 1:

**Source data 1.** Data for *Figure 1C*.

**Source data 2.** Data for *Figure 1E*.

**Source data 3.** Data for *Figure 1GH*.

**Figure supplement 1.** The number of L-plastin-positive cells colonizing developing retinas.

**Figure supplement 1—source data 1.** Data for *Figure 1—figure supplement 1B*.

**Figure supplement 2.** Spatio-temporal profile of microglial precursor colonization of developing retinas.

**Figure supplement 2—source data 1.** Data for *Figure 1—figure supplement 2*.

**Figure supplement 3.** Ocular microglial precursors are associated with newly differentiating amacrine cells.

**Figure supplement 4.** Peripheral macrophages of *Tg[EF1α:mCherry-zGem; mpeg1.1:EGFP]* transgenic embryos.

**Figure supplement 4—source data 1.** Data for *Figure 1—figure supplement 4D*.

---

except the OPL. Thus, microglial precursors enter the optic cup along the choroid fissure at 30 hpf, remain temporarily in the vitreous space between the lens and the retina, and then begin spreading into differentiating retinal tissue after 42 hpf.

Next, to evaluate the contribution of cell proliferation to the increasing number of ocular microglial precursors, we labeled ocular microglial precursors with markers of DNA replication. Here, we used a zebrafish transgenic line, *Tg[EF1α: mCherry-zGem]* that specifically marks proliferative cells in S and G2 phases (*Mochizuki et al., 2017*; *Mochizuki et al., 2014*). We combined this *Tg[EF1α: mCherry-zGem]* system with *Tg[mpeg1.1:EGFP]* to calculate the fraction of proliferative microglial precursors undergoing S phase (*Figure 1D* and *Video 1*). First, we observed mCherry-zGem; mpeg1.1:EGFP double-positive cells in the peripheral tissue (*Figure 1—figure supplement 4A-C*) and found that more than 60 % of mpeg1.1:EGFP-positive cells expressed mCherry-zGem (*Figure 1—figure supplement 4D*), confirming that this *Tg[EF1α: mCherry-zGem]* system works in early macrophages in zebrafish. However, in the retina, the fraction of mCherry-zGem; mpeg1.1:EGFP double-positive cells was less than 2 % of all microglial precursors from 32 to 54 hpf (*Figure 1E*). Furthermore, more than 80 % of mpeg1.1:EGFP-positive cells did not incorporate BrdU at 48 hpf (*Figure 1F–H*), suggesting that a majority of ocular microglial precursors do not undergo S phase. Thus, microglial colonization of the retina mostly depends on cell migration from outside the optic cup.

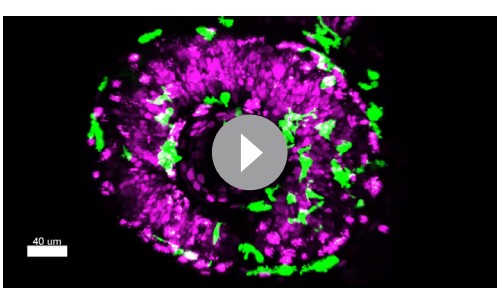

**Video 1.** 3D rendering of an eye of *Tg[EF1α: mCherry-zGem;mpeg1.1:EGFP]* zebrafish embryo at 42 hpf. mCherry-zGem signals indicate cells in S and G2 phase (magenta), whereas mpeg1.1:EGFP -positive cells are ocular microglial precursors (green). The fraction of mCherry-zGem; mpeg1.1:EGFP double-positive cells in mpeg1.1:EGFP-positive cells is very small, suggesting that almost all microglial precursors are in G1 phase.

https://elifesciences.org/articles/70550/figures#video1

## Embryonic microglial precursor migration into the retina depends on blood vessels

The zebrafish retina receives its blood supply from two blood vessel systems, intraocular hyaloid blood vessels encapsulating the lens (*Hartsock et al., 2014*) and superficial choroidal blood vessels (*Kaufman et al., 2015*). Developing hyaloid blood vessels start to enter the space between the lens and retina through the ventral fissure at 18–20 hpf. Loop formation occurs around the lens at 24–28 hpf, and a branched hyaloid network forms after 35 hpf (*Hartsock et al., 2014*). Our live imaging showed that microglial precursors enter the optic cup through the choroid fissure and remain temporarily in the vitreous space between the lens and the retina before they infiltrate the neural retina (*Figure 1—figure supplement 2*). Furthermore, microglial precursors start to enter the optic cup after loop formation of hyaloid blood vessels is completed, suggesting a guiding role of blood vessels in microglial precursor colonization of the optic cup. To confirm whether microglial precursors entering the ocular space are associated with developing hyaloid blood vessels, we conducted time-lapse imaging of *Tg[kdrl:EGFP; mfap4:tdTomato-CAAX]* transgenic embryos, which visualizes endothelial cells of blood vessels (*Jin et al., 2005*) and ocular microglial precursors, respectively. The first microglial precursor was always associated with ocular hyaloid blood vessels around 30 hpf (*Figure 2A*) and moved along blood vessel surfaces (*Video 2*), so it is very likely that microglial precursors use blood vessels as a scaffold to enter the vitreous space between the lens and the neural retina. Microglial precursors move along hyaloid blood vessels in the ventral fissure, gradually leave vessel surfaces, and invade the neural retina through the basement membrane (*Figure 2B*, *Figure 2—figure supplement 1*, and *Video 3*).

Troponin T2A (tnnt2a; silent heart) is specifically expressed in heart and is essential for heart contraction (*Sehnert et al., 2002*). In zebrafish brain and mouse retina, hemodynamics drive blood vessel pruning, and loss of blood circulation causes blood vessel regression (*Chen et al., 2012*; *Lobov et al., 2011*; *Yashiro et al., 2007*). To examine whether the entry of microglial precursors into retina is altered upon blood vessel regression, we blocked blood circulation by injecting morpholino antisense oligos against tnnt2a (tnnt2a MO). When blood circulation is inhibited, ocular hyaloid blood vessels do not develop fully and microglial precursors are less likely to be associated with these thin blood vessels (*Figure 2C*). The number of ocular microglial precursors was significantly reduced at 36 hpf (*Figure 2D*), showing that microglial colonization of the optic cup depends upon normal development of the blood vessel network. This is in contrast to the case of microglial colonization of zebrafish midbrain and optic tectum, which is independent of the blood vessel network (*Xu et al., 2016*). Indeed, we confirmed that the number of microglial precursors in the optic tectum was not significantly different between *tnnt2a* morphants and standard MO-injected embryos at 72 hpf, although microglial precursor colonization of the optic tectum was enhanced in *tnnt2a* morphants at 48 hpf (*Figure 2—figure supplement 2*).

Recent studies indicate that microglia facilitate ocular blood vessel development (*Checchin et al., 2006*; *Fantin et al., 2010*; *Rymo et al., 2011*), and that macrophages initiate a cell-death program in endothelial cells for blood vessel regression in developing mouse retina (*Lang and Bishop, 1993*; *Lobov et al., 2005*). However, we eliminated microglial precursors with morpholino antisense oligos against pu.1 (pu.1-MO) or *interferon regulatory factor 8* (*irf8*) mutation (*irf8* gene knockdown causes apoptosis of pu.1-positive myeloid cells) (*Shiau et al., 2015*), and confirmed that microglial precursor elimination did not affect hyaloid blood vessel formation in zebrafish at least by 48 hpf (*Figure 2—figure supplement 3*).

## Microglial precursors infiltrate the neural retina preferentially through the differentiating neurogenic area

In zebrafish, retinal neurogenesis occurs at the ventronasal retina adjacent to the optic stalk at 25 hpf and propagates into the entire region of the neural retina at 33 hpf (*Masai et al., 2000*). Microglial precursors start to migrate from the vitreous space into the neural retina after 42 hpf, when the earliest differentiating retinal neurons, RGCs, start to form the IPL (*Mumm et al., 2006*). To examine the role of retinal neurogenesis and RGC differentiation in microglia precursor infiltration of the neural retina, we used double transgenic lines, *Tg[EF1α: mCherry-zGem; mpeg1.1:EGFP]*, which enable us to examine the relationship between microglial precursor migration and retinal progenitor cells (*Mochizuki et al., 2014*). Live imaging of *Tg[EF1α: mCherry-zGem; mpeg1.1:EGFP]* retinas at 42 and 48 hpf

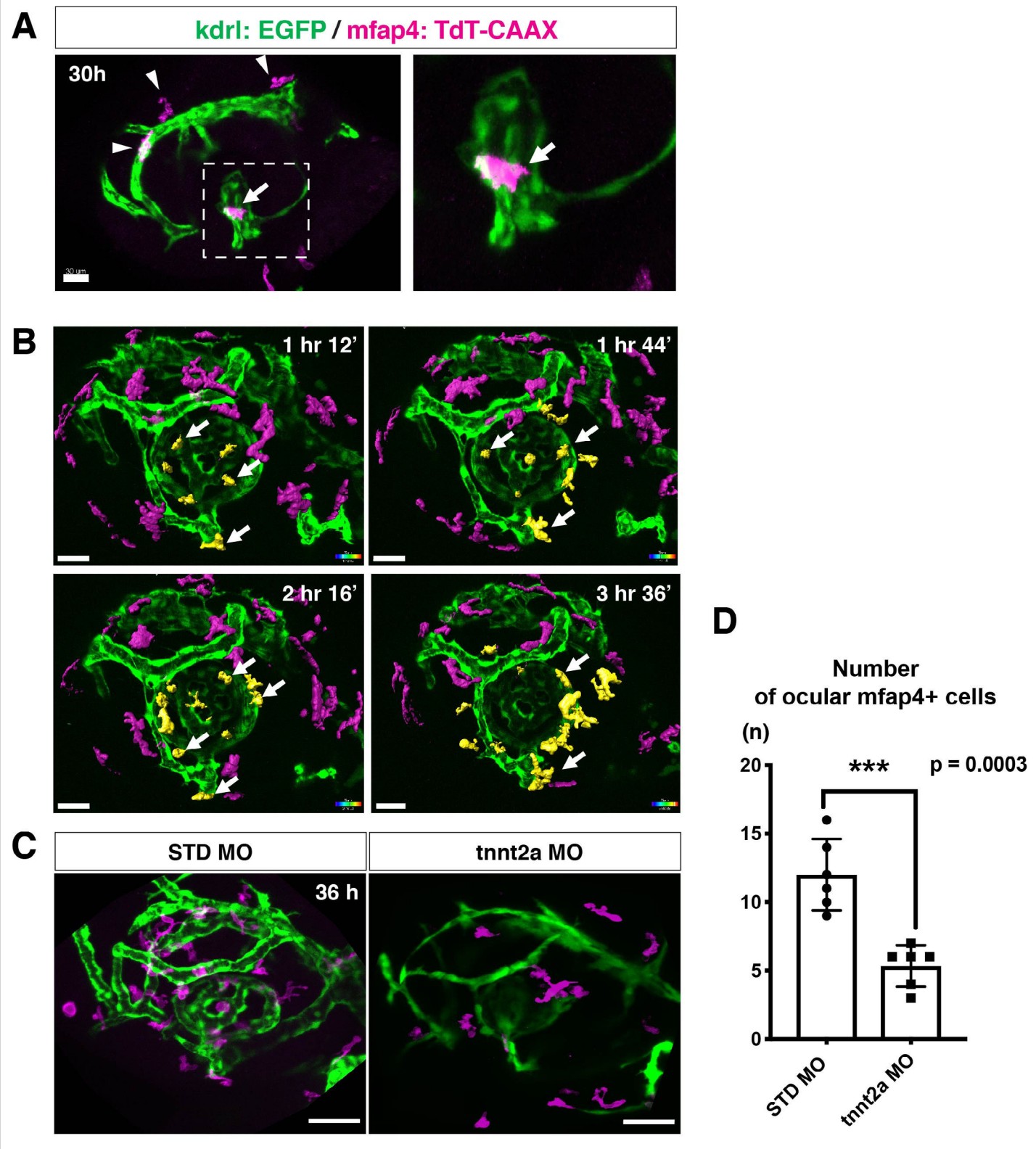

**Figure 2.** Microglial precursors migrate into the retina along blood vessels. (**A**) Live confocal images of *Tg[kdrl:EGFP; mfap4:tdTomato-CAAX]* retinas at 30 hpf. Microglial precursors and blood vessels are visualized using fluorescence of mfap4tdTomato-CAAX (magenta) and kdrl:EGFP (green), respectively. Higher magnification image of a dotted square in the left panel is shown in the right panel. The first microglial precursor (arrow) approaches along developing hyaloid blood vessels near the lens through the choroid fissure. Arrowheads indicate peripheral macrophages outside the optic cup.

*Figure 2 continued on next page*

*Figure 2 continued*

Scale bar: 30 µm. (**B**) Time-lapse 3D snapshots of *Tg[kdrl:EGFP; mfap4:tdTomato-CAAX]* eyes for around 3.5 hr after 32 hpf. Ocular microglial precursors and peripheral macrophages outside the optic cup are indicated as yellow- and magenta-colored, surface-rendered objects, respectively, which were prepared from the original scanning image (*Figure 2—figure supplement 1*). Ocular blood vessels are visualized in green. Microglia associated with hyaloid blood vessels around the lens (white arrows) gradually increase and infiltrate neurogenic retinal tissue (*Video 3*). Scale bar: 30 µm. (**C**) Live 3D images of eyes of *Tg[kdrl:EGFP; mfap4:tdTomato-CAAX]* embryos injected with standard MO and tnnt2a MO. kdrl:EGFP-positive blood vessels (green) are thinner in *tnnt2a* morphants. Scale bar: 50 µm. (**D**) Histogram of the number of intraocular microglial precursors in embryos injected with standard MO and tnnt2a MO. Bars and lines indicate means ± SD. ***p < 0.001.

The online version of this article includes the following figure supplement(s) for figure 2:

**Source data 1.** Data for *Figure 2D*.

**Figure supplement 1.** Time-lapse snapshots of *Tg[kdrl:EGFP; mfap4:tdTomato-CAAX]* heads for around 3.5 hr after 32 hpf.

**Figure supplement 2.** Microglial precursor colonization of the optic tectum does not depend on blood vessel formation.

**Figure supplement 2—source data 1.** Data for *Figure 2—figure supplement 2B*.

**Figure supplement 3.** Elimination of microglial precursors does not affect ocular blood vessel formation.

clearly showed that microglial precursors avoid mCherry-zGem-positive proliferating regions and are preferentially positioned in the region of mCherry-zGem-negative post-mitotic cells (*Figure 3A–B*, *Figure 3—figure supplement 1*). The fraction of microglial precursors that infiltrated mCherry-zGem-positive proliferating regions was 7.37 % at 42 hpf and 6.13 % at 48 hpf (*Figure 3C*), suggesting that >90% of microglial precursors infiltrate the retina through the mCherry-zGem-negative post-mitotic cell region. We also used another transgenic line *Tg[ath5:EGFP; mfap4:tdTomato-CAAX]*. In the *Tg[ath5:EGFP]* line, EGFP starts to be expressed in G2 phase of the final neurogenic cell division of retinal progenitor cells and is inherited by their daughter cells, which are negative for BrdU incorporation (*Poggi et al., 2005*; *Yamaguchi et al., 2010*), suggesting that ath5:EGFP specifically marks early differentiating retinal neurons. We conducted live imaging of *Tg[ath5:EGFP; mfap4:tdTomato-CAAX]* retinas at 36, 42, 48 hpf, and found that infiltration of mfap4-positive microglia preferentially occurs in the ath5:EGFP-positive region (*Figure 3D*). These data suggest that microglial precursors infiltrate the neural retina preferentially through the neurogenic area, raising the possibility that the neurogenic retinal region acts as a gateway through which microglial precursors move from the vitreous space into the neural retina.

Colonization of the optic tectum by microglial precursors depends on neuronal apoptosis in zebrafish (*Casano et al., 2016*; *Xu et al., 2016*). Therefore, it is still possible that microglial precursors preferentially infiltrate the neural retina through the neurogenic region, because of neuronal apoptosis. We inhibited retinal apoptosis by injecting morpholino antisense oligos against p53 (p53

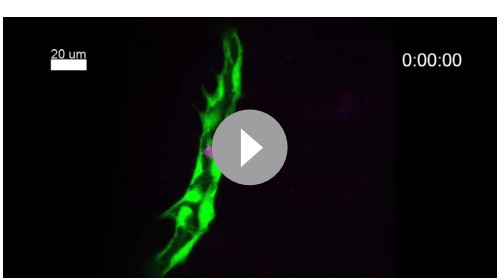

**Video 2.** Live imaging of *Tg[kdrl:EGFP; mfap4:tdTomato-CAAX]* embryo at 30 hpf. mfap4:tdTomato-CAAX-positive cells indicate microglial precursors (magenta), whereas kdrl:EGFP-positive cells indicate endothelial cells of blood vessels (green). Microglial precursors are moving on the surface of a developing superficial ocular blood vessel, suggesting that blood vessels act as scaffolds for migration of microglial precursors.
https://elifesciences.org/articles/70550/figures#video2

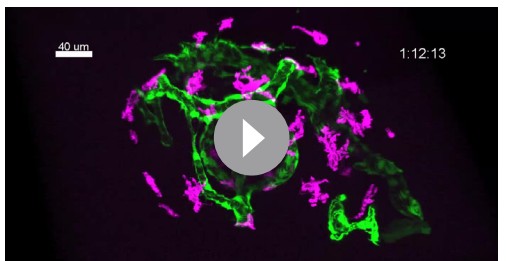

**Video 3.** Live imaging of *Tg[kdrl:EGFP; mfap4:tdTomato-CAAX]* embryos from 32 to 36 hpf. mfap4:tdTomato-CAAX-positive cells indicate microglial precursors (magenta), whereas kdrl:EGFP-positive cells indicate endothelial cells of blood vessels (green). Microglial precursors use blood vessels as scaffolds to migrate into the ocular space, and gradually invade the neural retina. Surface rendering indicates amoeboid microglial precursors, which are attached to the hyaloid loop around the lens and infiltrate the neural retina.
https://elifesciences.org/articles/70550/figures#video3

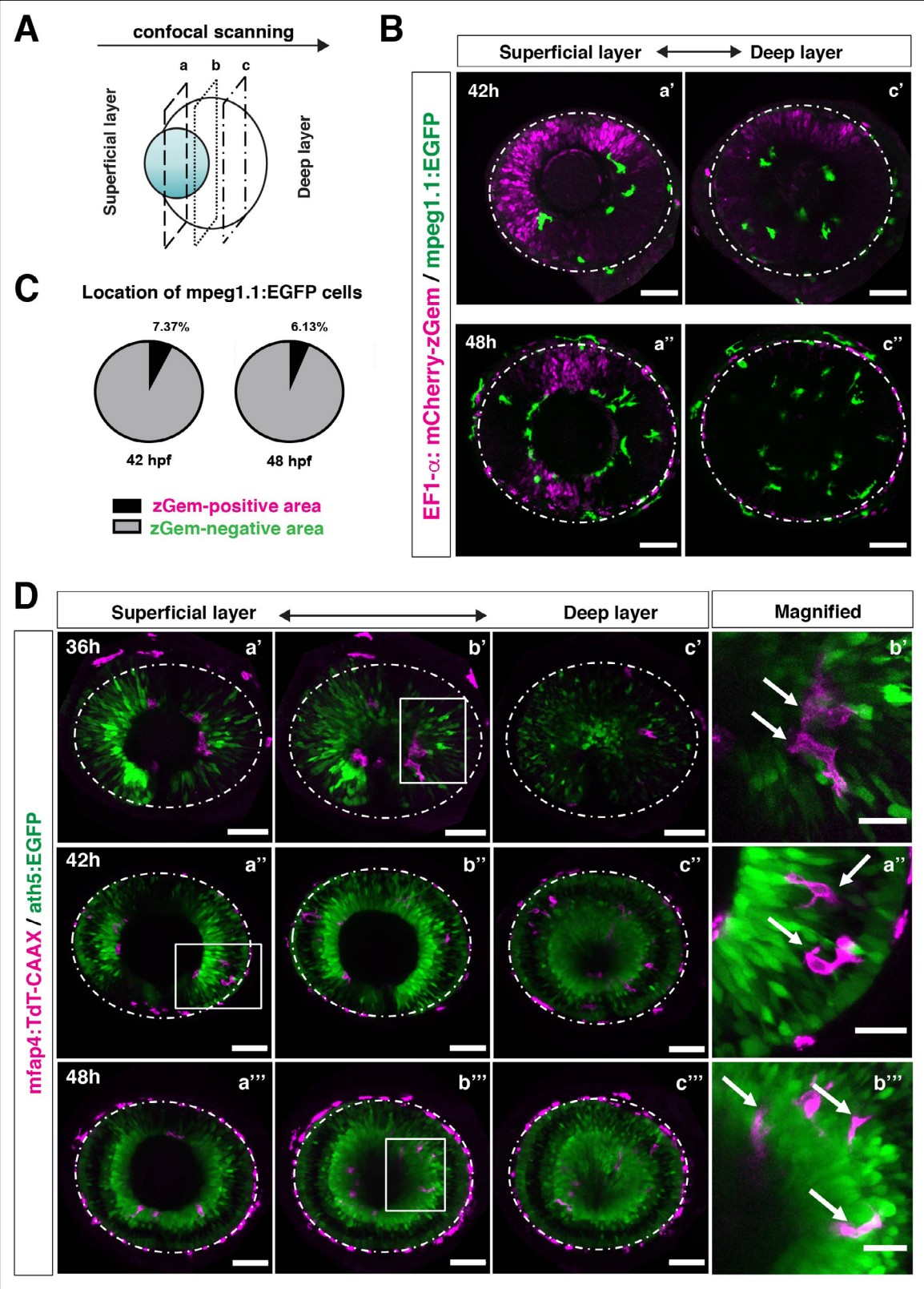

**Figure 3.** Microglial precursors infiltrate the retina through the neurogenic area. (**A**) Schematic drawing of confocal scanning planes (superficial, middle, and deep layers) in the optic cup shown in (**B**) and (**D**). (**B**) Live images of *Tg[EF1α:mCherry-zGem; mpeg1.1:EGFP]* retinas at 42 hpf (upper panels) and 48 hpf (lower panels). Two levels of confocal scanning planes are indicated as superficial (**a', a''**) and deep positions (**c', c''**). mpeg1.1:EGFP positive microglial precursors avoid mCherry-zGem positive proliferating retinal cell area. Scale bar: 50 μm. (**C**) Histogram of the fraction of microglial precursors

*Figure 3 continued on next page*

*Figure 3 continued*

associated with the mCherry-zGem-positive area (black) and the mCherry-zGem-negative area (grey). The fraction of microglial precursors associated with the mCherry-zGem-positive area is only 7.37 % at 42 hpf and 6.13 % at 48 hpf. Thus, more than 90 % of microglial precursors are located in the mCherry-zGem-negative retinal area. (**D**) Live images of *Tg[ath5:EGFP; mfap4:tdTomato-CAAX]* retinas at 36 (upper panels), 42 (middle panels) and 48 hpf (bottom panels). Three confocal scanning plane levels are indicated as superficial (**a'-a'''**), middle (**b'-b'''**), and deep (**c-c'''**). Dotted circles indicate the outline of the optic cup. The right-most column images indicate higher magnification images shown in the square of left panels. mfap4-positive microglia (magenta, arrows) are closely associated with ath5-positive neurogenic cells (green). Scale bar: 50 μm, except the right-most column images (Scale bar: 15 μm).

The online version of this article includes the following figure supplement(s) for figure 3:

**Source data 1.** Data for *Figure 3C*.

**Figure supplement 1.** Microglial precursors infiltrate the neural retina through the neurogenic region.

**Figure supplement 2.** p53 MO effectively inhibits retinal apoptosis in zebrafish.

**Figure supplement 2—source data 1.** Data for *Figure 3—figure supplement 2B*.

**Figure supplement 3.** Microglial precursor colonization depends on apoptosis in the optic tectum, but not in the retina.

**Figure supplement 3—source data 1.** Data for *Figure 3—figure supplement 3 B* D.

MO) and confirmed that p53 MO effectively suppresses retinal apoptosis at 24 and 36 hpf (*Figure 3—figure supplement 2*). However, the number of microglial precursors did not differ between *p53* morphant retinas and standard-MO-injected retinas at 48 hpf (*Figure 3—figure supplement 3A-B*), whereas the number of microglial precursors was significantly decreased in *p53* morphant optic tectum compared with standard-MO-injected optic tectum at 96 hpf (*Figure 3—figure supplement 3C-D*). Thus, in contrast to microglial colonization of the optic tectum, neuronal apoptosis is not the major cue for microglial precursor colonization of the retina, at least prior to 54 hpf.

## Neurogenesis acts as a gateway for microglial precursors to enter the retina

To confirm the possibility that the neurogenic retinal region functions as a gateway for microglial precursors to infiltrate the retina, we examined whether microglial precursor migration into the retina is compromised when retinal neurogenesis is affected. Previously, we found that histone genesis slowed in zebrafish *stem loop binding protein 1* (*slbp1*) mutants, leading to severe delays in retinal neurogenesis (*Imai et al., 2014*). Our bulk RNAseq analysis confirmed that retinal neurogenesis and subsequent neuronal differentiation were markedly delayed in zebrafish *slbp1* mutants (*Figure 4—figure supplement 1*), such that ath5 expression spread into the entire *slbp1* mutant retina only at 48 hpf, an event that occurs in wild-type retina at 33 hpf (*Imai et al., 2014*). We combined *slbp1* mutants with transgenic lines *Tg[ath5:EGFP; mfap4: tdTomato-CAAX]* and examined the number of ocular mfap4:tdTomato-CAAX-positive microglial precursors (*Figure 4A*, *Figure 4—figure supplements 2A and 3*). In 48-hpf *slbp1* mutant retinas, the number of ocular microglial precursors was 4.67 ± 2.42 (*Figure 4A and B*), which is similar to the number in wild-type retinas at 32 hpf (*Figure 1B*), whereas the number of ocular microglial precursors in wild-type siblings was 17.60 ± 5.13 at 48 hpf (*Figure 4A and B*). To confirm whether the *slbp1* mutation interferes with genesis of early macrophages, we examined peripheral mfap4[+] cells in the tail/trunk region of *slbp1* mutants and wild-type sibling embryos. There was no significant difference in mfap4[+] cells between *slbp1* mutants and wild-type siblings in the trunk/tail region (*Figure 4C and D*), indicating that the *slbp1* mutation does not influence early macrophage specification in zebrafish embryos. Although inhibition of retinal apoptosis by p53 MO does not influence microglial precursor colonization of the retina (*Figure 3—figure supplement 3A-B* ), we examined the level of retinal apoptosis in *slbp1* mutants. TUNEL revealed that apoptosis was increased in *slbp1* mutant retinas compared with wild-type sibling retinas (*Figure 4—figure supplement 4*). These data exclude the possibility that decreased retinal apoptosis affects microglial precursor colonization of the retina in *slbp1* mutants, and again confirm that neuronal apoptosis is not the major cue for microglial precursor colonization of the retina.

Mouse brain cortex colonization by microglia depends on the Cxcl12a-CxcR4 signaling axis (*Arnò et al., 2014*). We previously reported that *cxcl12a* expression is absent in the optic stalk of zebrafish *slbp1* mutants (*Imai et al., 2014*). To exclude the possibility that the absence of *cxcl12a* expression in the optic stalk affects microglial colonization of the retina in zebrafish *slbp1* mutants, we examined

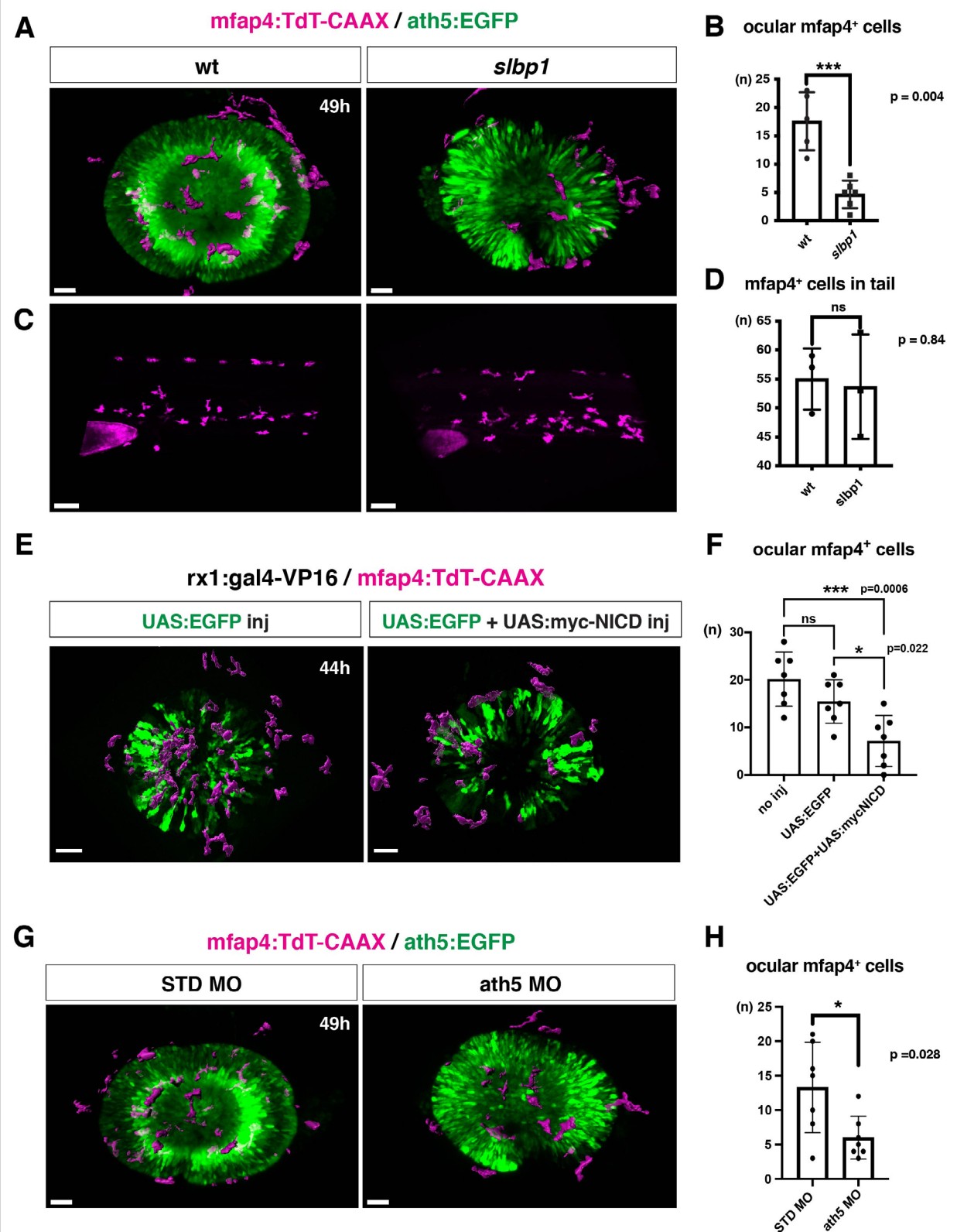

**Figure 4.** Microglial precursor infiltration into the retina depends on retinal neurogenesis. (**A**) Live 3D images of wild-type and *slbp1* mutant retinas with *Tg[mfap4:tdTomato-CAAX; ath5:EGFP]* at 49 hpf. Only mfap4:tdTomato-CAAX-positive ocular microglial precursors and peripheral macrophages are shown as surface-rendered objects. Original images are shown in *Figure 4—figure supplement 2A*. Scale bar: 30 μm. (**B**) Histogram of numbers of ocular microglial precursors in *slbp1* mutants and wild-type siblings. mfap4-positive microglial precursors are significantly fewer in *slbp1* mutants.

*Figure 4 continued on next page*

*Figure 4 continued*

Bars and lines indicate means ± SD. ***p < 0.001. (**C**) Live 3D images of wild-type and *slbp1* mutant trunk with *Tg[mfap4:tdTomato-CAAX; ath5:EGFP]* at 49 hpf. Scale bar: 70 µm. (**D**) Histogram of numbers of trunk macrophages in *slbp1* mutants and wild-type siblings. There is no significant difference in mfap4-positive macrophage number in trunks of *slbp1* mutants. Bars and lines indicate means ± SD. (**E**) Live 3D images of retinas of *Tg[rx1:gal4-VP16; mfap4:tdTomato-CAAX]* embryos injected with one DNA construct encoding *UAS:EGFP* (left) or two DNA constructs encoding *UAS:EGFP; UAS:myc-tagged NICD* (right) at 44 hpf. Only mfap4:tdTomato-CAAX-positive ocular microglial precursors and peripheral macrophages are shown as surface-rendered objects. Original images are shown in *Figure 4—figure supplement 2B*. Scale bar: 30 µm. (**F**) Histogram of numbers of ocular microglial precursors in *rx1:gal4-VP16; UAS:EGFP* expressed and *rx1:gal4-VP16; UAS:EGFP; UAS:myc-NICD* expressed wild-type retinas. mfap4-positive microglia are significantly decreased in *myc-NICD* expressed retinas, compared with non-injection control and *EGFP* expressed control retinas. Bars and lines indicate means ± SD. *P < 0.05, ***P < 0.001. (**G**) Live 3D images of standard MO- and ath5 MO-injected retinas of *Tg[mfap4:tdTomato-CAAX; ath5:EGFP]* embryos at 49 hpf. Only mfap4:tdTomato-CAAX-positive ocular microglial precursors and peripheral macrophages are shown as surface-rendered objects. Original images are shown in *Figure 4—figure supplement 2C*. Scale bar: 30 µm. (**H**) Histogram of numbers of ocular microglial precursors in standard MO and ath5 MO-injected wild-type retinas. mfap4-positive microglial precursors are significantly less numerous in *ath5* morphant retinas. Bars and lines indicate means ± SD. *p < 0.05.

The online version of this article includes the following figure supplement(s) for figure 4:

**Source data 1.** Data for *Figure 4B*.

**Source data 2.** Data for *Figure 4D*.

**Source data 3.** Data for *Figure 4F*.

**Source data 4.** Data for *Figure 4H*.

**Figure supplement 1.** Retinal neurogenesis and cell differentiation are markedly delayed in zebrafish *slbp1* mutants.

**Figure supplement 2.** Original scanning images of *Figure 4A, E and G*.

**Figure supplement 3.** Extraction process of ocular microglial precursors from original 3D scanning images.

**Figure supplement 4.** Retinal apoptosis is increased in zebrafish *slbp1* mutants.

**Figure supplement 4—source data 1.** Data for *Figure 4—figure supplement 4B*.

**Figure supplement 5.** Microglial precursor colonization of the retina is independent of the Cxcl12a signaling pathway.

**Figure supplement 5—source data 1.** Data for *Figure 4—figure supplement 5D*.

**Figure supplement 6.** Elimination of microglial precursors does not affect retinal cell differentiation.

**Figure supplement 7.** Overexpression of NICD suppresses retinal neurogenesis in zebrafish.

**Figure supplement 7—source data 1.** Data for *Figure 4—figure supplement 7C*.

**Figure supplement 8.** Microglial precursor colonization of the retina is affected in *ath5* morphants.

zebrafish *cxcl12a* morphants. Injection of *cxcl12a*-MO at 500 µM, which effectively induces RGC axon trajectory defects reported in zebrafish *odysseys* mutants carrying mutations in *cxcl12a* receptor, *cxcr4b* (*Li et al., 2005*), did not affect the number of ocular microglial precursors (*Figure 4—figure supplement 5*). Thus, Cxcl12a-CxcR4 signaling is not involved in microglial colonization defects in *slbp1* mutants. We also confirmed that elimination of microglial precursors with pu.1 MO did not affect the rate of retinal neurogenesis or cell differentiation by 72 hpf (*Figure 4—figure supplement 6*).

We previously showed that overexpression of Notch1 intracellular domain (NICD) suppresses retinal neurogenesis in zebrafish (*Yamaguchi et al., 2005*). We confirmed that overexpression of NICD suppresses retinal neurogenesis in zebrafish by injecting a DNA expression construct encoding UAS:myc-NICD (*Scheer and Campos-Ortega, 1999*) into *Tg[hsp:gal4; ath5:EGFP]* double transgenic embryos (*Figure 4—figure supplement 7*). Next, we examined whether microglial precursor infiltration of the retina is compromised in retinas overexpressing NICD. We established a zebrafish transgenic line, *Tg[rx1:gal4-VP16]*, which expresses Gal4-VP16 under control of a retinal progenitor-specific promoter *rx1* (*Chuang et al., 1999*), and then injected two DNA expression constructs encoding UAS:EGFP (*Köster and Fraser, 2001*) and UAS:myc-NICD into *Tg[rx1:gal4-VP16; mfap4:tdTomato-CAAX]* double-transgenic embryos. Embryos injected with only the DNA construct of UAS:EGFP served as a positive control. We selected embryos in which EGFP was expressed in most retinal cells at 24 hpf and used them for further analysis. The number of ocular microglial precursors was significantly reduced in embryos overexpressing NICD and EGFP, compared with control embryos overexpressing EGFP, at 44 hpf (*Figure 4E and F*, *Figure 4—figure supplements 2B and 3*). These data support

the possibility that the retinal neurogenic region functions as a gateway for microglia to infiltrate the retina.

The blockade of retinal neurogenesis delays differentiation of the first-born retinal cell-type, RGCs. To examine whether blockade of RGC differentiation affects microglial precursor colonization of the neural retina, we applied an antisense morpholino against ath5 (known as atoh7) (ath5 MO). As with the zebrafish *ath5* mutant, *lakritz* (*Kay et al., 2001*), RGC differentiation was specifically inhibited in *ath5* morphant retinas (*Figure 4—figure supplement 8A*, B). In *ath5* morphants, the timing of the first appearance of microglial precursors in the ocular vitreous space was not altered, but the number of ocular microglial precursors was significantly decreased at 49 hpf (*Figure 4G and H*, and *Figure 4— figure supplements 2C and 3* and 8 C), suggesting that RGC differentiation or RGC-mediated IPL formation is required for microglial precursor infiltration into the neural retina.

## Microglial precursors preferentially associate with neurogenic retinal columns

To determine whether microglia precursors have greater affinity for differentiating neurons than for retinal progenitor cells, we carried out two sets of experiments. First, we conducted cell transplant experiments using a wild-type donor line and an *slbp1* mutant recipient line carrying *Tg[mfap4:tdTomato-CAAX]*. Wild-type donor cells were transplanted into *slbp1* mutant recipient embryos at the blastula stage. We selected *slbp1* mutant and wild-type sibling embryos in which wild-type, donor retinal cell columns were introduced in a mosaic manner at 48 hpf (*Figure 5A*). Host microglial precursors and donor retinal cells were visualized with mfap4:tdTomato-CAAX and Alexa-488 Dextran, respectively. In *slbp1* mutant host retinas, microglial precursors were likely to be associated with donor wild-type retinal columns more frequently than in wild-type host retinas (*Figure 5B*). To analyze these data statistically, we compared eyes in which wild-type donors were transplanted into wild-type hosts with those in which wild-type donors were transplanted into *slbp1* mutant hosts (*Figure 5—figure supplement 1A-B*). The fraction of microglial precursors associated with donor wild-type retinal columns in total ocular microglial precursors was significantly higher in *slbp1* mutant host retinas than in wild-type sibling host retinas at 48 hpf (*Figure 5C*), suggesting that microglial precursors are more attracted by wild-type donor neurogenic retinal columns than surrounding *slbp1* mutant proliferative retinal cells. Since the fraction of microglial precursors associated with donor retinal columns in total microglial precursors may depend on the number of donor retinal columns incorporated into host retinas, we next estimated trapping efficiency of microglial precursors per donor column by dividing the fraction of microglial precursors associated with donor columns with the transplanted donor column number in each eye (*Figure 5—figure supplement 1B*). Trapping efficiency of microglial precursors per donor column was significantly higher in *slbp1* mutant host retinas than in wild-type sibling host retinas (*Figure 5D*), suggesting that microglial precursors are preferentially associated with donor-derived wild-type retinal cells than with host-derived *slbp1* mutant retinal cells.

Second, we injected two DNA constructs encoding UAS:EGFP and UAS:mycNICD into *Tg[hsp:gal4; mfap4:tdTomato-CAAX]* double-transgenic wild-type embryos. Two rounds of heat-shock treatment at 18 and 30 hpf induced expression of NICD and EGFP in a mosaic manner in the retina (*Figure 5E*). We examined the fraction of *mfap4:tdTomato-CAAX*-positive microglial precursors associated with EGFP-expressing retinal columns in the total number of *mfap4:tdTomato-CAAX*-positive microglial precursors (*Figure 5F*). This fraction was significantly lower in retinas overexpressing NICD and EGFP than in control retinas overexpressing only EGFP (*Figure 5G*, *Figure 5—figure supplement 1C*). We also confirmed that trapping efficiency of *mfap4:tdTomato-CAAX*-positive microglial precursors per EGFP-positive retinal column was significantly lower in retinas overexpressing NICD and EGFP than in retinas overexpressing only EGFP (*Figure 5H*, *Figure 5—figure supplement 1C*). Thus, microglial precursors are less attracted by retinal columns in which neurogenesis is arrested. Taken together, these data suggest that microglial precursors preferentially associate with neurogenic retinal columns as opposed to proliferative retinal columns.

## IL34 is involved in microglial precursor colonization of the retina

Recently, it was reported that microglial colonization of zebrafish brain, including retina, depends on CSF-R, and that one of the CSF-R ligands, IL34, dominates this process (*Wu et al., 2018*). In adult mouse retina, RGCs express IL34, which attracts one subset of microglia and retains them around the

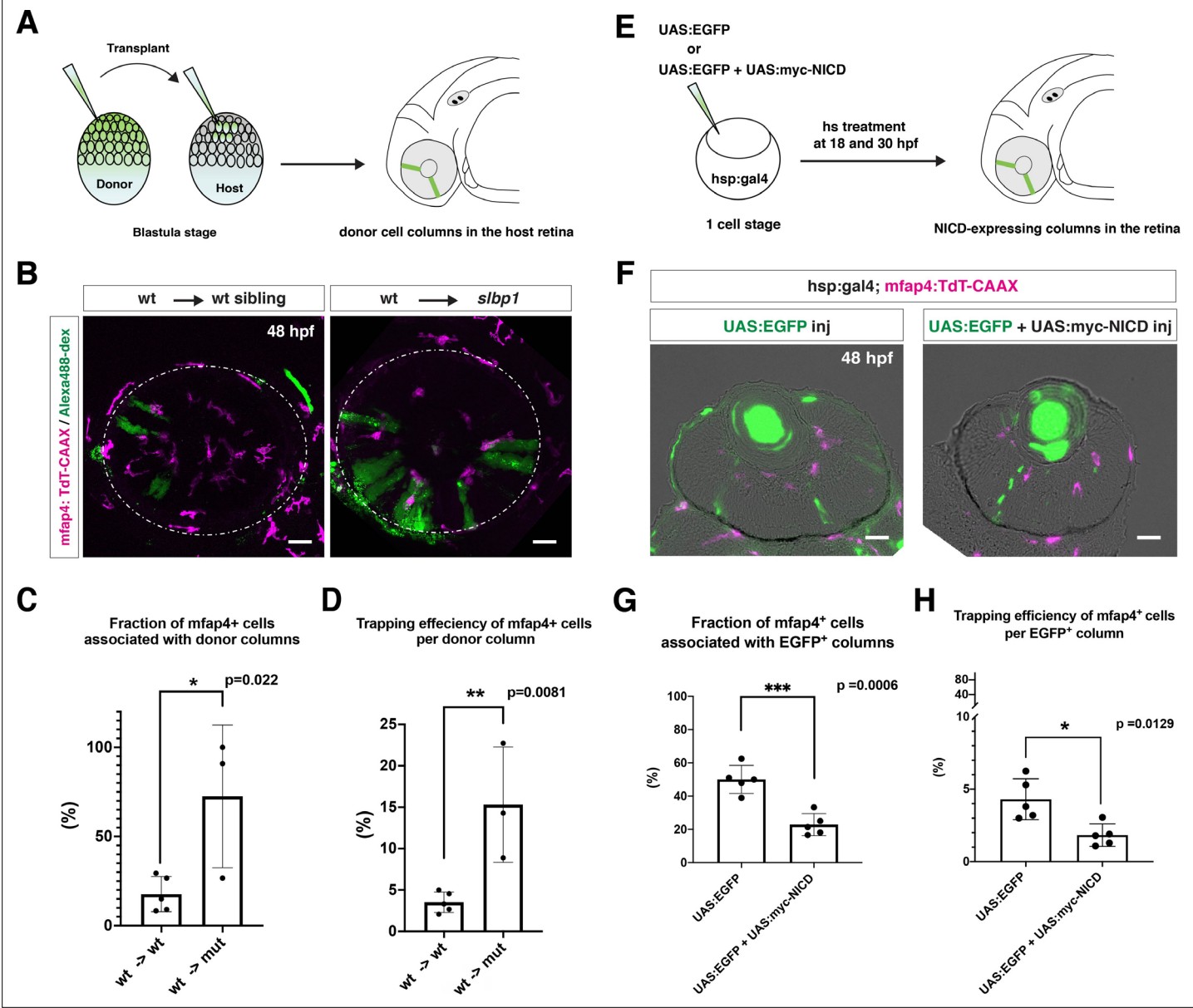

**Figure 5.** Microglial precursors are preferentially associated with neurogenic retinal columns. (**A**) Schematic drawing of cell transplantation experiments. Wild-type donor embryos are labeled with Alexa-448-dextran and transplanted into *slbp1* mutant recipient embryos at blastula stage. In *slbp1* mutant recipient embryos, transplanted wild-type donor cells form retinal cell columns. The host *slbp1* mutant line is combined with *Tg[mfap4:tdTomato-CAAX]*, to investigate whether mfap4-positive microglial precursors (magenta) infiltrate the neural retina preferentially through Alexa-448-dextran-labeled, wild-type donor columns (green) in *slbp1* mutant recipient embryos. (**B**) Live images of *slbp1* mutant retinas with transplanted wild-type donor retinal cell columns at 48 hpf. Donor wild-type retinal cell columns are labeled with Alexa-488 dextran (green). Host microglial precursors are visualized with the transgene *Tg[mfap4:tdTomato-CAAX]* (magenta). Dotted circles indicate the outline of the optic cup. Many microglial precursors are associated with wild-type donor retinal columns in *slbp1* mutant host retinas (right panel), compared with wild-type sibling host retinas (left panel). Scale bar: 30 μm (**C**) The fraction of mfap4-positive microglial precursors associated with donor transplanted retinal cell columns versus the total number of microglial precursors in the optic cup. The average fraction of mfap4-positive cells associated with donor retinal cell columns is significantly higher in *slbp1* mutant host retinas than in wild-type host retinas. Bars and lines indicate means ± SD. *p < 0.05. (**D**) The trapping efficiency of mfap4-positive microglial precursors per donor column. The average trapping efficiency is significantly higher in *slbp1* mutant host retinas than in wild-type host retinas, suggesting higher affinity of microglial precursors for neurogenic retinal cells. Bars and lines indicate means ± SD. **p < 0.01. (**E**) Schematic drawing of mosaic expression of NICD in retinas. A mixture of UAS:EGFP and UAS-myc-NICD plasmids was injected into fertilized eggs of the *Tg[hsp:gal4; mfap4:tdTomato]* transgenic line, which were treated by heat shock at 18 and 30 hpf. At 48 hpf, embryos were fixed to prepare serial retinal sections for imaging analysis. (**F**) Confocal scanning of retinal sections of *Tg[hsp:gal4; mfap4:tdTomato]* transgenic embryos injected with plasmids encoding UAS:EGFP or UAS:EGFP+ UAS:myc-NICD. Scale bar: 30 μm. (**G**) The fraction of mfap4-positive microglial precursors associated with EGFP-expressing

*Figure 5 continued on next page*

*Figure 5 continued*

retinal cell columns versus the total number of microglial precursors in the optic cup. The average fraction of mfap4-positive cells associated with EGFP-positive retinal columns is significantly lower in retinas injected with UAS:EGFP+ UAS:myc-NICD than with only UAS:EGFP control. Bars and lines indicate means ± SD. ***p < 0.005. (**H**) The trapping efficiency of mfap4-positive microglial precursors per EGFP-expressing retinal cell columns. The average trapping efficiency is significantly lower in retinas injected with UAS:EGFP+ UAS:myc-NICD than with only UAS:EGFP control, suggesting less affinity of microglial precursors for proliferative NICD-expressing retinal cells. Bars and lines indicate means ± SD. *p < 0.05.

The online version of this article includes the following figure supplement(s) for figure 5:

**Source data 1.** Data for *Figure 5CDGH*.

**Figure supplement 1.** Calculation of trapping efficiency of microglial precursors by retinal columns.

IPL niche (*O'Koren et al., 2019*). First, we confirmed that retinal cell differentiation proceeds normally until 72 hpf in zebrafish *il34* mutants, although pyknotic nuclei were stochastically observed in RGC and amacrine cell layers (*Figure 6—figure supplement 1*). Next, we examined microglial precursor colonization of the retina. The number of ocular microglial precursors was significantly lower in *il34* homozygous mutants than in wild-type siblings at 34 hpf (*Figure 6—figure supplement 2*) and 48 hpf (*Figure 6A and B*). Thus, consistent with the previous report (*Wu et al., 2018*), IL34 is required for microglial precursor colonization of the retina in zebrafish. However, *il34* mRNA expression is comparable in *slbp1* mutant heads and wild-type sibling heads at 48 hpf (*Figure 6—figure supplement 3*), suggesting that *il34* mRNA expression is not linked to retinal neurogenesis. Since the number of ocular microglial precursors in *il34* homozygous mutants was zero at 34 hpf (*Figure 6—figure supplement 2*) and no more than two, if any, at 48 hpf (*Figure 6B*), it is likely that Csf1r-il34 signaling promotes microglial precursor movement from yolk to the optic cup upstream of the blood-vessel-mediated guidance mechanism (*Figure 6C*).

## Discussion

In zebrafish, primitive microglia originate from the RBI, which is a hematopoietic tissue equivalent to mouse yolk sac, whereas definitive microglia are generated from hematopoietic stem cells that are specified in the VDA (*Ferrero et al., 2018*; *Xu et al., 2015*). Primitive and definitive waves of hematopoiesis generate embryonic and adult microglia, respectively. Using zebrafish as an animal model, several groups investigated microglial colonization from the periphery into developing brain, especially the optic tectum, which is part of the midbrain (*Casano et al., 2016*; *Herbomel et al., 2001*; *Svahn et al., 2013*; *Wu et al., 2018*; *Xu et al., 2016*). Colonization of the optic tectum by microglial precursors depends on neuronal apoptosis, probably through attraction by an apoptotic cell-secreted phospholipid, lysophosphatidylcholine (LPC) (*Casano et al., 2016*; *Xu et al., 2016*). In addition, microglial colonization of brain is CSF receptor-dependent (*Herbomel et al., 2001*; *Wu et al., 2018*). In mice, microglial colonization of brain requires functional blood circulation (*Ginhoux et al., 2010*). However, in zebrafish, microglial colonization of the optic tectum is independent of blood circulation (*Xu et al., 2016*). A series of elegant studies revealed the molecular network that promotes microglial colonization of the midbrain. However, it remains to be seen whether this mechanism fully explains colonization of other brain regions by microglial precursors. In this study, we focused on zebrafish retina and investigated the mechanism that regulates migration of embryonic microglial precursors into developing retina.

We first conducted live imaging of zebrafish microglial precursors from 24 to 54 hpf. Microglial precursors progressively increase in number during embryonic development. Interestingly, almost all microglial precursors enter the optic cup through the choroid fissure. However, peripheral macrophages located in the mesenchymal region between the eye and the brain did not enter the optic cup across the ciliary marginal zone. This may be consistent with the observation that these peripheral macrophages never enter the retina following rod cell death (*White et al., 2017*), suggesting a functional difference between peripheral macrophages and ocular microglia. Next, we found that the majority of ocular microglial precursors do not undergo S phase and are probably in G1 phase. Thus, the increase of ocular microglial precursors is due to migration from outside the eye. In developing mouse retina, microglial precursors appear from the vitreous area near the optic disk at E11.5, progressively increase in number, and then infiltrate the neural retina. These retinal microglia were

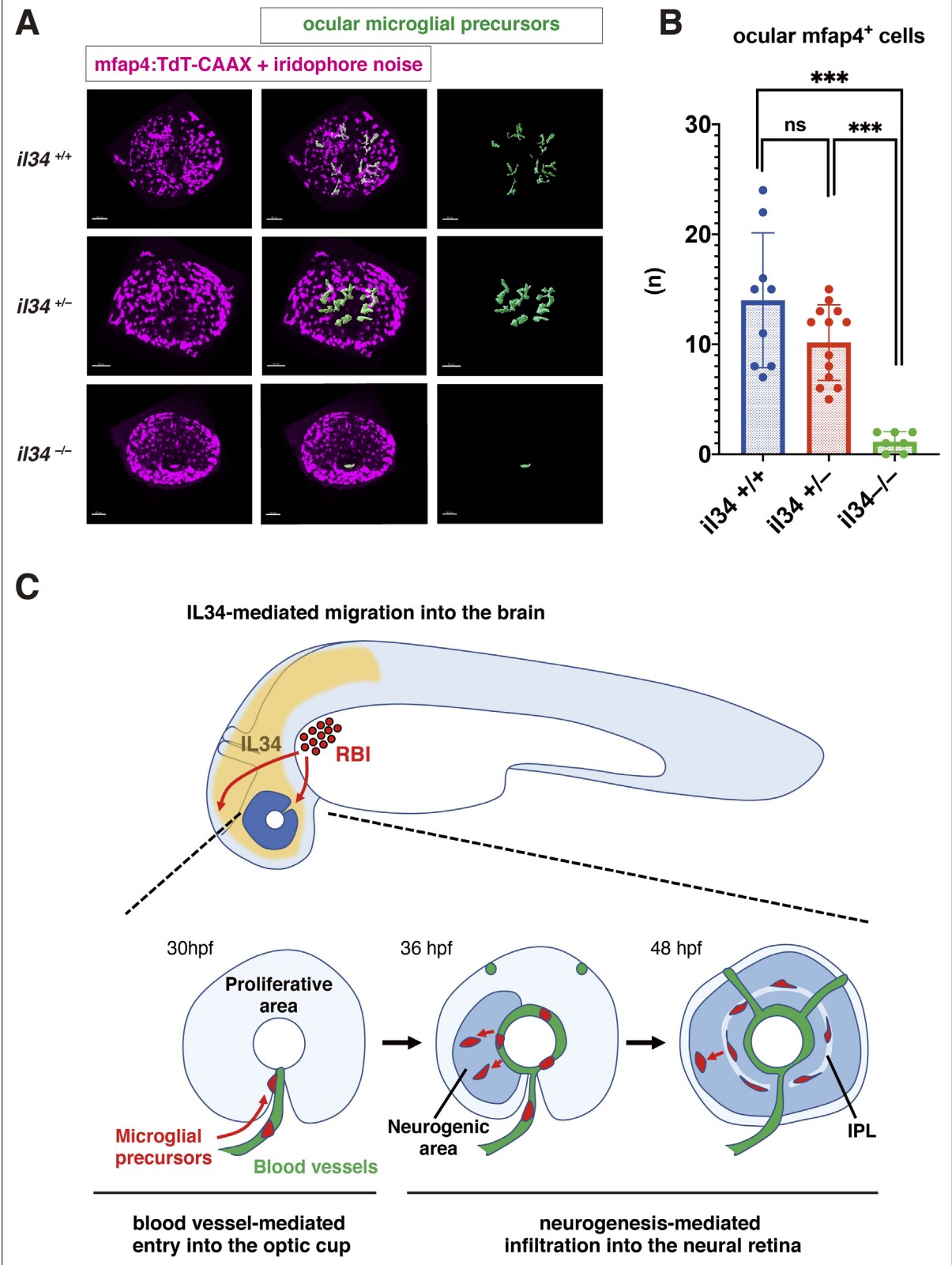

**Figure 6.** IL34 is required for colonization of the optic cup by microglial precursors. (**A**) Confocal 3D scanning of 48 hpf wild-type, *il34* heterozygous and homozygous mutant retinas carrying the *Tg[mfap4:tdTomato-CAAX]* transgene. At 48 hpf, iridophores start to differentiate around the optic cup, which causes a noise signal (magenta) in confocal scanning. Using the surface-rendering tool of Imaris software (Bitplane), we eliminated iridophore-derived noise and extracted mfap4:tdTomato-CAAX signals from ocular microglial precursors (green) (see the legend of *Figure 4—figure supplement 3*). Scale

*Figure 6 continued on next page*

*Figure 6 continued*

bar: 50 µm. (**B**) Histogram of numbers of ocular microglial precursors in wild-type, *il34* heterozygous and homozygous mutant retinas at 48 hpf. The number of ocular microglial precursors is almost zero, and very few, if any (one or two), in *il34* homozygous mutants, indicating that ocular microglial precursors are significantly reduced in *il34* homozygous mutants. The number of ocular microglial precursors is mildly reduced in *il34* heterozygous mutants, but does not differ significantly from that of wild-type siblings. Bars and lines indicate means ± SD. ***$p < 0.005$. (**C**) A possible model of the guidance mechanism of microglial precursor into zebrafish retina. IL34 is involved in movement of microglial precursors toward the brain. Microglial precursors continue into the optic cup along blood vessels, and subsequently infiltrate the neural retina through the neurogenic area.

The online version of this article includes the following figure supplement(s) for figure 6:

**Source data 1.** Data for *Figure 6B*.

**Figure supplement 1.** Retinal cell differentiation normally proceeds in *il34* mutants.

**Figure supplement 2.** Microglial precursor colonization of the optic cup in *il34* mutants.

**Figure supplement 2—source data 1.** Data for *Figure 6—figure supplement 2B*.

**Figure supplement 3.** *il34* mRNA expression is comparable between wild-type sibling and *slbp1* mutant heads.

**Figure supplement 3—source data 1.** Data for *Figure 6—figure supplement 3*.

---

also negative for a proliferative marker, Ki67 (*Santos et al., 2008*), suggesting that mouse embryonic retinal microglia are also non-proliferative.

Another interesting finding is that entry of microglial precursors into the optic cup through the choroid fissure depends on ocular blood vessels. We observed that migrating microglial precursors are closely associated with hyaloid blood vessels after loop formation. These microglial precursors pass along these vessels, which traverse the choroid fissure and surround the posterior region of the lens. Furthermore, the number of ocular microglial precursors was reduced when blood circulation was blocked. Since inhibition of blood circulation compromises the structural integrity of blood vessels in zebrafish, we conclude that ocular blood vessel formation is required for microglial precursor entry into the optic cup through the choroid fissure. One possibility is that blood vessels function as a path upon which microglial precursors enter the optic cup. Membrane proteins or extracellular matrix proteins on blood endothelial cells may facilitate the association of microglial precursors with blood vessel surfaces. Alternatively, substances that attract microglial precursors may be released from hyaloid blood endothelial cells. Previous studies on human and murine microglia demonstrated that microglial colonization of the retina takes place prior to retinal vascularization, and that microglia facilitate ocular blood vessel development (*Checchin et al., 2006*; *Fantin et al., 2010*; *Rymo et al., 2011*). Macrophages initiate endothelial cell death for blood vessel regression in developing mouse retina (*Lang and Bishop, 1993*; *Lobov et al., 2005*). However, in contrast to mammals, elimination of microglia by pu.1 MO or *irf8* mutation did not affect ocular blood vessel formation in zebrafish, suggesting that microglia do not regulate ocular blood vessel formation in zebrafish. Interestingly, classic histological studies on mouse retinas showed that early emerging ocular microglia are associated with the hyaloid artery (*Hume et al., 1983*; *Santos et al., 2008*), which is located in the vitreous area and regresses in later stages before retinal vasculature formation (*Ito and Yoshioka, 1999*). Thus, further investigation will be necessary to determine whether the hyaloid artery guides microglial precursors into the optic cup in vertebrate species such as mice. In zebrafish, colonization of the optic tectum by microglia is independent of blood circulation (*Xu et al., 2016*). We confirmed that the number of microglia in the optic tectum did not differ between *tnnt2a* morphants and control embryos at 72 hpf; however, microglial colonization of the optic tectum was enhanced and microglial shape was round rather ramified in *tnnt2a* morphants at 48 hpf. Further study will be necessary to clarify the role of blood circulation in microglial colonization of the optic tectum.

After 42 hpf, microglial precursors detach from hyaloid blood vessels and start to infiltrate the neural retina. Interestingly, we found that more than 90 % of microglial precursors enter the neural retina through the neurogenic area. Indeed, the number of microglial precursors is reduced in *slbp1* mutant retinas and NICD-overexpressing retinas, in both of which retinal neurogenesis is severely delayed. Furthermore, we conducted two sets of experiments: the first was cell transplantation from wild-type donor cells into *slbp1* mutant host retinas, which introduced neurogenic wild-type retinal cell columns in proliferative *slbp1* mutant retinas, and the second was overexpression of NICD in wild-type retina, which introduced proliferative retinal cell columns in neurogenic retinas. Consistently, in both cases, microglial precursors were preferentially associated with neurogenic retinal cell columns.

Thus, neurogenesis is required for infiltration of microglial precursors into the neural retina after 42 hpf. We observed that the number of microglial precursors is diminished in *ath5* morphant retinas, suggesting that RGCs are required for infiltration of microglial precursors into the neural retina. There are at least three possible mechanisms for this infiltration. First, the basal region of retinal progenitor cells may function as a physical barrier that inhibits microglial precursor infiltration of the neural retina. Second, microglial precursors may be attracted to surfaces of differentiating retinal neurons or RGCs. Third, differentiating retinal neurons or RGCs may release a specific attractant for microglia. There are several candidate molecules that suggest the third possibility. In adult mice, RGCs express IL34, which attracts microglia and retains them around the IPL niche (*O'Koren et al., 2019*). Indeed, microglial colonization of zebrafish brain depends on CSF-R, and one of the CSF-R ligands, IL34, dominates this process (*Wu et al., 2018*). We confirmed that microglial precursor colonization of retina is severely affected in *il34* mutants. However, *il34* mRNA expression is comparable in *slbp1* mutants and their wild-type siblings, suggesting that IL34 is not linked to neurogenesis-mediated microglial precursor infiltration. Rather, the number of ocular microglial precursors in *il34* mutants was almost zero at 34 and 48 hpf, so it is very likely that Csf1r-il34 signaling initiates microglial precursor movement from yolk toward brain and retina, followed by blood vessel- and neurogenesis-mediated guidance.

It was reported that apoptosis attracts microglia in zebrafish developing brain (*Casano et al., 2016*; *Xu et al., 2016*). However, microglial precursor colonization of the retina is normal in zebrafish *p53* morphants, suggesting that apoptosis does not promote microglial precursor colonization of the retina. Why are microglial precursors insensitive to retinal apoptosis? We found that apoptosis is enhanced in zebrafish *slbp1* mutant retinas, in which microglial precursor colonization is severely affected due to a delay of retinal neurogenesis. It is likely that spontaneous apoptotic cells fail to be eliminated because of the reduced number of microglial precursors in *slbp1* mutant retinas; however, interestingly, these increased dead cells did not promote microglial precursor infiltration into *slbp1* mutant retinas, suggesting that neurogenesis primarily opens the gate through which microglial precursors enter the neural retina. Since retinal neurogenesis normally occurs from 24 to 48 hpf in zebrafish, microglial precursors could not be attracted by apoptosis without the infiltration path opened by neurogenesis before 48 hpf. Further studies will be necessary to unveil the molecular mechanism underlying microglial infiltration into neural retina.

In summary, there are three mechanisms for microglial colonization of developing zebrafish retina (*Figure 6C*). IL34-CSF-R signaling initiates microglial precursor movement from yolk toward brain and retina. Microglial precursors further use ocular hyaloid blood vessels as a pathway to enter the optic cup and then infiltrate the neural retina preferentially through the neurogenic region. In the future, it remains to identify molecules involved in blood-vessel- and neurogenesis-mediated guidance mechanisms, and to assess whether these mechanisms are used for microglial colonization of other brain regions in other vertebrate species.

## Materials and methods

### Fish strains

Zebrafish (*Danio rerio*) were maintained using standard procedures (*Westerfield, 1993*). RIKEN wako (RW) was used as a wild-type strain for mutagenesis (*Masai et al., 2003*). *slbp1*[rw440] (*Imai et al., 2014*), *irf8*[st96] (*Shiau et al., 2015*) and *il34*[hkz11] (*Wu et al., 2018*) were used. Transgenic lines *Tg[ath5:EGFP]*[rw021] were used to monitor *ath5* gene expression (*Masai et al., 2005*). *Tg[EF1α:mCherry-zGem]*[pki011] (*Mochizuki et al., 2014*) was used for visualization of cell-cycle phases. *Tg[mfap4:tdTomato-CAAX]*[pki058] and *Tg[mpeg1.1:EGFP]*[pki053] were used to visualize microglial precursors. *Tg[kdrl:EGFP]*[s843Tg] was employed to visualize blood vessels (*Jin et al., 2005*). *Tg[hsp:gal4]*[kca4] (*Scheer et al., 2002*) and *Tg[rx1:gal4-VP16]*[pki065] were used for UAS-mediated expression of target genes. For confocal scanning, embryos were incubated with 0.003 % phenyltiourea (PTU) (Nacalai tesque, 27429–22) to prevent melanophore pigmentation. The zebrafish pigmentation mutant, *roy orbison* (*roy*) (*D'Agati et al., 2017*) was used to remove iridophores.

## Establishment of *Tg[mpeg1.1:EGFP] and Tg[mfap4:tdTomato-CAAX]* transgenic lines

The DNA construct encoding mpeg1.1:EGFP was kindly provided by Dr. Graham Lieschke and we are indebted to Dr. David Tobin for the construct encoding mfap4:tdTomato-CAAX. These DNA constructs were injected into fertilized eggs with Tol2 transposase mRNA, to establish transgenic lines, *Tg[mpeg1.1:EGFP]* and *Tg[mfap4:tdTomato-CAAX]* in our lab.

## Histology

Plastic sectioning and immunolabeling of cryosections were carried out as previously described (*Masai et al., 2003*). Anti-GFP (Themo Fisher Scientific, A11122), anti-myc-tag (Invitrogen, R950-25), zn5 (Oregon Monoclonal Bank) and zpr1 (Oregon Monoclonal Bank) antibodies were used at 1:200; 1:250, 1:100, and 1:100 dilutions, respectively. For detection of BrdU incorporation, BrdU (Nacalai, tesque, 05650–95) was applied to 52-hpf embryos, chased for 2 hr at 28.5 °C and fixed with 4 % paraformaldehyde (PFA). Labeling of retinal sections with anti-BrdU antibody (BioRad, MCA2060) was carried out as previously described (*Yamaguchi et al., 2005*). TUNEL was performed using an In Situ Cell Death Detection Kit (Roche, 11684795910). Bodipy-ceramide (Thermo Fisher Scientific, B22650) was applied to visualize retinal layers as previously described (*Masai et al., 2003*). Nuclear staining was performed using 1 nM TOPRO3 (Thermo Fisher Scientific, T3605).

## Morpholino

Morpholino antisense oligos were designed as shown below.

> tnnt2a MO: 5'-CATGTTTGCTCTGATCTGACACGCA-3' (*Sehnert et al., 2002*)
> p53 MO: 5'-GCGCCATTGCTTTGCAAGAATTG-3' (*Langheinrich et al., 2002*)
> cxcl12a MO: 5'-ACTTTGAGATCCATGTTTGCAGTG-3' (*Li et al., 2005*)
> pu.1 MO: 5'-GATATACTGATACTCCATTGGTGGT-3' (*Rhodes et al., 2005*)
> ath5 MO: 5'-TTCATGGCTCTTCAAAAAAGTCTCC-3'
> Standard MO: 5'-CCTCTTACCTCAGTTACAATTTATA-3'

Morpholino antisense oligos were injected into fertilized eggs at 500 µM for tnnt2a MO and cxcl12a MO; 250 µM for ath5 MO and pu.1 MO and 100 µM for p53 MO. The same concentration was used for Standard MO in each MO experiment.

## Cell transplantation

Cell transplantation was performed as previously described (*Masai et al., 2003*). Wild-type zygotes were injected with Alexa-488 dextran (Thermo Fisher Scientific, D22910) and used for donor embryos. *slbp1* mutant embryos carrying *Tg[mfap4:tdTomato-CAAX]* were used as host embryos. Host embryos carrying donor retinal cells were selected by observing Alexa 488 fluorescence at 24 hpf. *slbp1* mutant and wild-type sibling embryos were sorted based on the *slbp1* mutant morphological phenotype at 48 hpf and used for live imaging. After confocal images were obtained, the number of ocular mfap4-positive microglial precursors associated with Alexa-488 dextran-labeled donor transplanted retinal columns was counted. The fraction of ocular mfap4-positive microglial precursors associated with donor transplanted retinal columns in total ocular microglial precursors was calculated. The trapping efficiency of ocular mfap4-positive microglial precursors per transplanted donor retinal column was calculated using the total number of donor transplanted retinal columns in the retina. Detailed information on each transplanted eye is shown in *Figure 5—figure supplement 1A-B*.

## Live imaging and analyses

Transgene lines *Tg[mpeg1.1:EGFP]* or *Tg[mfap4:tdTomato-CAAX]*, and *Tg[kdrl:EGFP]*, were used for time-lapse imaging of microglial precursors and blood vessels. 3D confocal images were obtained using a confocal LSM, LSM710 (Zeiss) or an FV3000RS (Olympus), and analyzed using ImageJ (2.0.0-rc-69/1.52 p) and Imaris software (ver.9.1.2 Bitplane). The DNA construct encoding Ptf1a:EGFP was used for visualizing amacrine cells or their progenitors (*Jusuf and Harris, 2009*).

## RNA extraction

Heads of 48 hpf wild-type sibling and *slbp1* mutant embryos were dissected and transferred to 100 µL Sepasol (Nacalai tesque, 09379) on ice. Heads were then homogenized using a hand homogenizer (~20 pulses). Twenty µL $CHCl_3$ were then added to samples and mixed gently. After centrifugation at 15,000 g for 15 min, the aqueous phase was collected and mixed with 100 µL isopropanol. One µL of RNase-free glycogen (Nacalai tesque 11170–11, 20 mg/mL) was added to all samples to increase the yield. After incubating at room temperature for 10 min, samples were centrifuged at 15,000 g at 4 °C for 15 min. Supernatant was removed and the pellet was washed three times with 500 µL of 75 % ethanol and centrifuged at 8000 g at 4 °C. The pellet was then resuspended in a desired amount of nuclease-free water and stored at –80 °C. RNA concentration and purity of samples were determined using a Nanodrop.

## RNA sequencing and analysis

RNA samples with RIN >7 were subjected to paired-end sequencing using an Illumina HiSeq4000. First, a quality check was performed using FastQC and read trimming was done with Trimomatic (*Bolger et al., 2014*). PRINSEQ lite (*Schmieder and Edwards, 2011*) was used for PolyA trimming and quality filtering. Trimmed sequences were then mapped to the zebrafish reference genome (GRCz11) using hisat2.1.0 (*Kim et al., 2019*) and mapped reads are counted using featureCounts (*Liao et al., 2014*). With the R package, EdgeR (*Robinson et al., 2010*), differentially expressed genes with $Log_2FC > |2|$ and FDR values < 0.01 were extracted. EnhancedVolcano package (*Blighe et al., 2018*) was used to draw volcano plots. A heat map was generated with the pheatmap package (*Kolde, 2019*).

## Evaluation of *Il34* mRNA expression by semi-quantitative PCR

Extracted RNA from 48-hpf wild-type sibling and *slbp1* mutant heads was used to prepare cDNA, using ReverTra Ace qPCR RT master mix with gDNA remover (Toyobo, FSQ-301). The expression level of *il34* mRNA was evaluated with quantitative PCR using the primers below. mRNA of cytoplasmic actin β2, namely *actb2* (ZFIN), was used for normalization.

> Forward primer for *il34* mRNA: 5'-TGGTCCAGTCCGAATGCT-3'.
> Reserve primer for *il34* mRNA: 5'-GCTGCACTACTGCACACTGG-3'.
> Forward primer for *actb2* mRNA: 5'-TGTCTTCCCATCCATCGTG-3'.
> Reserve primer for *actb2* mRNA: 5'-TGTCTTCCCATCCATCGTG-3'.

## Mosaic expression of NICD in retinal cells using *Tg[rx1:gal4-VP16]* and *Tg[hsp:gal4]* transgenic lines

The DNA fragment that covers a 2892 bp genomic region upstream from the start codon of *rx1* cDNA (*Chuang et al., 1999*), was amplified by PCR and inserted between *XhoI* and *BamHI* sites of the Tol2 base expression vector, pT2AL200R150G (*Urasaki et al., 2006*). Next, DNA fragments encoding *gal4-VP16* (*Köster and Fraser, 2001*) were further inserted between *BamHI* and *ClaI* sites of pT2AL200R150G to fuse the *rx1* promoter. The plasmid was injected with Tol2 transposase mRNA into fertilized eggs of the UAS:EGFP transgenic line to establish a transgenic line, *Tg[rx1:gal4-VP16]^pki065*. A mixture of plasmids of UAS:EGFP (*Köster and Fraser, 2001*) and UAS:myc-NICD (*Scheer and Campos-Ortega, 1999*) (each 10 ng/µL) were injected into fertilized eggs of the *Tg[mfap4:tdTomato-CAAX; rx1:gal4-VP16]* or *Tg[mfap4:tdTomato-CAAX; hsp:gal4]* transgenic line. In the case of the *Tg[mfap4:tdTomato-CAAX; hsp:gal4]* transgenic line, two rounds of heat shock at 37 °C for 1 hr were applied at 18 and 30 hpf. Embryos expressing EGFP in the optic cup were selected at 24 hpf, fixed with PFA at 48 hpf and used to prepare serial retinal sections for imaging analysis. After confocal images were obtained, the number of ocular mfap4-positive microglial precursors associated with EGFP-expressing columns was counted. The fraction of ocular mfap4-positive microglial precursors associated with EGFP-expressing columns in total microglial precursors was calculated and the trapping efficiency of ocular mfap4-positive microglial precursors per EGFP-expressing column was calculated using the total number of EGFP-expressing columns in the retina. Detailed information on each injected eye is shown in *Figure 5—figure supplement 1C*. To confirm that NICD inhibits retinal neurogenesis, UAS:myc-NICD or UAS:mCherry (each 10 ng/µL) was injected into zebrafish transgenic embryos *Tg[ath5:EGFP; hsp:gal4]*. Three rounds of heat shock at 37 °C for 1 hr were applied at 18,

24, and 30 hpf. Embryos were fixed at 36 hpf and labeled with anti-myc tag antibody to visualize myc-NICD expressing retinal cells with Alexa-543-conjugated secondary antibody. Whole retinas were used for confocal scanning with an FV3000RS (Olympus). Controls were UAS:mCherry-injected samples and used directly for live confocal scanning. Confocal 3D retinal images were used to count the number of ath5:EGFP-positive and negative retinal columns in myc-NICD or mCherry expressing retinal columns from five independent embryos.

## Evaluation of microglial precursor colonization of the retina in *Il34* mutants

The *il34*[hkz11] allele (*Wu et al., 2018*) was combined with the *Tg[mfap4:tdTomato]* transgenic line and used for analysis. Embryos were generated by pair-wise crosses between heterozygous mutant male and female fish, and maintained with N-phenyl thiourea (PTU)-containing water to prevent melanophore pigmentation. Whole retinas of 19 embryo at 34 hpf and 29 embryos at 48 hpf were scanned with confocal microscopy, using an LSM710 (Zeiss) or an FV3000RS (Olympus). Embryos were fixed with 4 % PFA and used for genotyping. A DNA fragment containing the 4-base deletion mutation of the *il34*[hkz11] allele was amplified by PCR and sequenced to determine genotypes. Primers used for PCR amplification and sequencing are below.

> Forward primer for PCR: 5'-TGCAATTAAACAGCCAATGTG-3'.
> Reverse primer for PCR: 5'-CTGAGTCACAGCCCTCAAATC-3'.
> Forward primer for sequencing: 5'-CCATTTGTTTTTACCTGACCAAA-3'.
> Reverse primer for sequencing: 5'-GCTAATTGGTGTGGGACGTT-3'.

Using the surface rendering tool of Imaris software (Bitplane, ver.9.1.2), we eliminated signals of iridophore-derived noise or peripheral macrophages around the optic cup and extracted only ocular microglial precursors. The number of ocular microglial precursors was counted in each retina and compared between genotypes.

## Statistical analysis

Statistical analyses were performed using GraphPad Prism version 8.2.1. Statistical significance was determined using two-tailed unpaired Student's t-tests for *Figure 1G*; *Figure 2D*; *Figure 4B,D,H*; *Figure 5C,D,G,H*; *Figure 1—figure supplement 2*; *Figure 2—figure supplement 2B*; *Figure 3—figure supplement 3B-D*; *Figure 4—figure supplement 4B*; *Figure 4—figure supplement 5D*; *Figure 6—figure supplement 3*, Tukey's multiple comparison test for *Figure 4F*; *Figure 6B*; *Figure 6—figure supplement 2B*, and Bonferroni's multiple comparison test for *Figure 3—figure supplement 2B*. Chi square tests were used for *Figure 4—figure supplement 7C*. Detailed information on each dataset is provided in Excel files in Raw data.

## Acknowledgements

We thank Graham Lieschke for DNA constructs encoding mepg1.1:EGFP, David Tobin for DNA constructs encoding mfap4:tdTomato-CAAX, Francesco Argenton for DNA construct encoding Ptf1a:EGFP, William Talbot for zebrafish *irf8* mutant line, Zilong Wen for zebrafish *il34* mutant line, and José Campos-Ortega for DNA constructs encoding UAS-myc tagged NICD. We also thank lab members, especially Yuko Nishiwaki, Yuki Takeuchi, Yutaka Kojima, Jeff Liner, Mamoru Fujiwara, and Tetsuya Harakuni for supporting experiments. We thank Steven D Aird for editing the manuscript.

## Additional information

### Funding

| Funder | Grant reference number | Author |
|---|---|---|
| Okinawa Institute of Science and Technology Graduate University | | Ichiro Masai |

| Funder | Grant reference number | Author |
|---|---|---|

The funders had no role in study design, data collection and interpretation, or the decision to submit the work for publication.

## Author contributions

Nishtha Ranawat, Conceptualization, Data curation, Formal analysis, Investigation, Methodology, Project administration, Resources, Software, Validation, Visualization, Writing – original draft, Writing – review and editing; Ichiro Masai, Conceptualization, Data curation, Formal analysis, Funding acquisition, Investigation, Methodology, Project administration, Resources, Supervision, Validation, Visualization, Writing – original draft, Writing – review and editing

### Author ORCIDs

Nishtha Ranawat ⓘ http://orcid.org/0000-0001-9577-1171
Ichiro Masai ⓘ http://orcid.org/0000-0002-6626-6595

### Ethics

Ethics statementAll zebrafish experiments were performed in accordance with the Animal Care and Use Program of Okinawa Institute of Science and Technology Graduate University (OIST), Japan, which is based on the Guide for the Care and Use of Laboratory Animals by the National Research Council of the National Academies. The OIST animal care facility has been accredited by the Association for Assessment and Accreditation of Laboratory Animal Care (AAALAC International). All experimental protocols were approved by the OIST Institutional Animal Care and Use Committee.

### Decision letter and Author response

Decision letter https://doi.org/10.7554/eLife.70550.sa1
Author response https://doi.org/10.7554/eLife.70550.sa2

---

## Additional files

### Supplementary files

• Transparent reporting form

### Data availability

Raw RNA-seq dataset of slbp1 mutant and wild-type sibling is available at Gene Expression Omnibus (GSE144517).

The following dataset was generated:

| Author(s) | Year | Dataset title | Dataset URL | Database and Identifier |
|---|---|---|---|---|
| Nishtha R, Ichiro M | 2020 | Comparsion of transcriptome between slbp1 mutant and wildtype sibling | https://www.ncbi.nlm.nih.gov/geo/query/acc.cgi?acc=GSE144517 | NCBI Gene Expression Omnibus, GSE144517 |

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

# Appendix 1

### Appendix 1—key resources table

| Reagent type (species) or resource | Designation | Source or reference | Identifiers | Additional information |
|---|---|---|---|---|
| Strain, strain background (zebrafish, *Danio rerio*) | Okinawa wild type | PMID:28196805 | | |
| Strain, strain background (zebrafish, *Danio rerio*) | RIKEN wild type | PMID:12702661 | ZFIN: ZDB-GENO-070802–4 | https://shigen.nig.ac.jp/zebra/ |
| Genetic reagent (zebrafish, *Danio rerio*) | *slbp1*$^{rw440}$ | PMID:25106852 | ZFIN: ZDB-ALT-150115–5 | |
| Genetic reagent (zebrafish, *Danio rerio*) | *irf8*$^{st96}$ | PMID:25615614 | ZFIN: ZDB-ALT-150504–8 | Dr. William Talbot (Stanford University) |
| Genetic reagent (zebrafish, *Danio rerio*) | *il34*$^{hkz11}$ | PMID:30205037 | ZFIN: ZDB-ALT-181210–3 | Dr. Zilong Wen (The Hong Kong University of Science and Technology) |
| Genetic reagent (zebrafish, *Danio rerio*) | *Roy orbison* | PMID:28760346 | ZFIN: ZDB-ALT-980203–444 | |
| Genetic reagent (zebrafish, *Danio rerio*) | *Tg[ath5:EGFP]*$^{rw021}$ | PMID:12702661 | ZFIN: ZDB-ALT-050627–2 | |
| Genetic reagent (zebrafish, *Danio rerio*) | *Tg[EF1a:mCherry-zGem]*$^{oki011}$ | PMID:25260917 | ZFIN: ZDB-ALT-150128–2 | |
| Genetic reagent (zebrafish, *Danio rerio*) | *Tg[mpeg1.1:EGFP]*$^{oki053}$ | This paper | | See "Materials and Methods" |
| Genetic reagent (zebrafish, *Danio rerio*) | *Tg[mfap4:tdTomato]*$^{oki083}$ | This paper | | See "Materials and Methods" |
| Genetic reagent (zebrafish, *Danio rerio*) | *Tg[kdrl:EGFP]*$^{s843Tg}$ | PMID:16251212 | ZFIN: ZDB-ALT-050916–14 | ZIRC |
| Genetic reagent (zebrafish, *Danio rerio*) | *Tg[hsp:gal4]*$^{kca4}$ | PMID:11850174 | ZFIN: ZDB-ALT-020918–6 | Reugels/Campos-Ortega lab (Köln University) |
| Genetic reagent (zebrafish, *Danio rerio*) | *Tg[rx1:gal4-VP16]*$^{oki065}$ | This paper | | See "Materials and Methods" |
| Antibody | zn5 (mouse monoclonal) | ZIRC | ZFIN: ZDB-ATB-081002–19 | IHC (1:100) |
| Antibody | zpr1 (mouse monoclonal) | ZIRC | ZFIN: ZDB-ATB-081002–43 | IHC (1:100) |
| Antibody | Anti-GFP (rabbit polyclonal) | Thermo Fisher Scientific | Cat# A11122 | IHC (1:200) |
| Antibody | Anti-myc tag (mouse monoclonal) | Invitrogen | Cat# R950-25 | IHC (1:250) |
| Antibody | Anti-BrdU (rat monoclonal) | BioRad | Cat# MCA2060 | Monoclonal (BU1/75(ICR1)) IHC (1:200) |
| Recombinant DNA reagent | pT2AL200R150G (plasmid) | PMID:16959904 | | Dr. Koichi Kawakami (Institute of Genetics) |
| Recombinant DNA reagent | UAS:EGFP (plasmid) | PMID:11336499 | | 10 ng/µL for injection |
| Recombinant DNA reagent | UAS:mCherry (plasmid) | This paper | | 10 ng/µL for injection |
| Recombinant DNA reagent | UAS:myc-NICD (plasmid) | PMID:10072782 | | Reugels/Campos-Ortega lab (Köln University) 10 ng/µL for injection |
| Recombinant DNA reagent | Ptf1a:EGFP (plasmid) | PMID:19732413 | | Dr. Francesco Argenton (University of Padova) 10 ng/µL for injection |
| Sequence-based reagent | tnnt2a MO | PMID:11967535 | Morpholino antisense oligos | 5'-CATGTTTGCTCTGATCTGACACGCA-3' Use at 500 µM |
| Sequence-based reagent | p53 MO | PMID:12477391 | Morpholino antisense oligos | 5'-GCGCCATTGCTTTGCAAGAATTG-3' Use at 100 µM |

*Appendix 1 Continued on next page*

*Appendix 1 Continued*

| Reagent type (species) or resource | Designation | Source or reference | Identifiers | Additional information |
|---|---|---|---|---|
| Sequence-based reagent | cxcl12a MO | PMID:15716407 | Morpholino antisense oligos | 5'-ACTTTGAGATCCATGTTTGCAGTG-3' Use at 500 µM |
| Sequence-based reagent | pu.1 MO | PMID:15621533 | Morpholino antisense oligos | 5'-GATATACTGATACTCCATTGGTGGT-3' Use at 250 µM |
| Sequence-based reagent | ath5 MO | This paper | Morpholino antisense oligos | 5'-TTCATGGCTCTTCAAAAAAGTCTCC-3' Use at 250 µM |
| Sequence-based reagent | standard MO | other | Morpholino antisense oligos | 5'-CCTCTTACCTCAGTTACAATTTATA-3' Use at the same concentration for each MO experiments |
| Sequence-based reagent | il34 qPCR primer forward | This paper | PCR primers | 5'- TGGTCCAGTCCGAATGCT-3' |
| Sequence-based reagent | il34 qPCR primer reverse | This paper | PCR primers | 5'- GCTGCACTACTGCACACTGG –3' |
| Sequence-based reagent | actb2 qPCR primer forward | This paper | PCR primers | 5'- TGTCTTCCCATCCATCGTG –3' |
| Sequence-based reagent | actb2 qPCR primer reverse | This paper | PCR primers | 5'- TGTCTTCCCATCCATCGTG-3' |
| Sequence-based reagent | il34 genotyping primer forward | This paper | PCR primers | 5'-TGCAATTAAACAGCCAATGTG-3' |
| Sequence-based reagent | il34 genotyping primer reverse | This paper | PCR primers | 5'-CTGAGTCACAGCCCTCAAATC-3' |
| Sequence-based reagent | il34 sequencing primer forward | This paper | PCR primers | 5'-CCATTTGTTTTTACCTGACCAAA-3' |
| Sequence-based reagent | il34 g sequencing primer reverse | This paper | PCR primers | 5'-GCTAATTGGTGTGGGACGTT-3' |
| Commercial assay or kit | In Situ Cell Death Detection Kit, Fluorescein | Roche | Cat# 11684795910 | |
| Commercial assay or kit | Sepasol-RNA/ Super G | Nacalai tesque | Cat# 09379 | |
| Commercial assay or kit | ReverTra Ace aPCR master mix with gDNA remover | Toyobo | Cat# FSQ-301 | |
| Chemical compound, drug | BrdU | Nacalai tesque | Cat# 05650–95 | |
| Chemical compound, drug | Bodipy ceramide | Thermo Fisher Scientific | Cat# B22650 | |
| Chemical compound, drug | TO-PRO–3 Iodide (642/661) | Thermo Fisher Scientific | Cat# T3605 | |
| Chemical compound, drug | N-Phenyl thiourea (PTU) | Nacalai tesque | Cat# 27429–22 | |
| Chemical compound, drug | Dextran, Alexia Flour-488 | Thermo Fisher Scientific | Cat# D22910 | |
| Software, algorithm | GraphPad Prism | GraphPad Software | Ver 8.2.1 | https://www.graphpad.com/scientific-software/prism/ |
| Software, algorithm | IMARIS | Bitplane | Ver 9.1.2 | http://www.bitplane.com/imaris; RRID:SCR_007370 |
| Software, algorithm | Image-J | NIH | 2.0.0-rc-69/1.52 p | |
| Software, algorithm | Trimomatic | PMID:24695404 | v0.39 | http://www.usadellab.org/cms/?page=trimmomatic |
| Software, algorithm | PRINSEQ lite | PMID:21278185 | v0.20.4 | http://prinseq.sourceforge.net/ |
| Software, algorithm | HISAT2 | PMID:25751142 | v2.1.0 | https://github.com/DaehwanKimLab/hisat2 |
| Software, algorithm | featureCounts | PMID:24227677 | Packaged ub Subread v1.5.2 | http://subread.sourceforge.net/ |

*Appendix 1 Continued on next page*

*Appendix 1 Continued*

| Reagent type (species) or resource | Designation | Source or reference | Identifiers | Additional information |
|---|---|---|---|---|
| Software, algorithm | EdgeR | PMID:19910308 | v3.13 | https://bioconductor.org/packages/release/bioc/html/edgeR.html |
| Software, algorithm | EnhancedVolcano package | *Blighe et al., 2018* | v1.6.0 | https://github.com/kevinblighe/EnhancedVolcano |
| Software, algorithm | pheatmap package | other | v1.10.12 | https://cran.r-project.org/web/packages/pheatmap/index.html |

