## [Editor Report]

The authors have addressed the remaining concerns raised by reviewers and the revised manuscript has been strengthened with revisions to the text and figures.

This manuscript will be of use to developmental neurobiologists and provides new insight on the mechanisms and microglia-vascular interactions and microglial colonization of the zebrafish retina.

---

## [Decision Letter]

[Editors’ note: the authors submitted for reconsideration following the decision after peer review. What follows is the decision letter after the first round of review.]

Thank you for choosing to send your work, "Mechanisms underlying microglial colonization of developing neural retina in zebrafish", for consideration at *eLife*. Your article has been reviewed by 2 peer reviewers and the evaluation has been overseen by a Senior Editor and a Reviewing Editor. Although the work is of interest, we regret to inform you that the findings at this stage are too preliminary for further consideration at *eLife*.

The authors address an important question and gap in knowledge on how microglia colonize the mammalian retina. The findings are interesting, however the findings do not support their main conclusion that microglia enter the retina through a neurogenic niche. A major weakness is the manipulations used to disrupt the neurogenic niche are global and thus it is not clear whether or how microglia and/or other tissues are impacted by these manipulations. The study lacks some key controls making several experiments difficult to interpret. Moreover, the tools and markers used to label macrophages and microglia (including proliferating microglia) need more characterization and validation. There were several other specific concerns as outlined below.

*Reviewer #2:*

Summary:

The authors use the zebrafish model to understand how microglia colonize the retina. Through time lapse imaging experiments the authors show that microglia enter the retinal space through hyaloid blood vessels. These microglia do not express zGEM, a proxy for proliferating cells. Once in the blood vessels, myeloid precursors enter into the retina. The entrance of these microglia occurs in areas of the retina that are not undergoing active proliferation. The authors claim that such results indicate neurogenic properties then drive colonization. Using multiple approaches to disrupt retinal neurogenesis, the authors show that microglia colonization can be disrupted. Overall, the paper presents interesting and important concepts on how microglia colonize the retina. A basic understanding and characterization of microglia retinal colonization is needed. However, some of the conclusions in the work are overstated based on the current data. In particular, the supporting data and manipulations to disrupt the neurogenic niche are performed globally, leaving the possibility that the manipulations could also be altering microglia directly or indirectly through other tissues. A number of conclusions are drawn from experiments that did not provide an effect. Such interpretations need strong positive controls, of which are missing from the current manuscript. The conclusions stated in the paper will need additional experiments. The manuscript could be vastly improved through revision; please consider these comments for improving the paper.

1. The authors use the term microglial precursors often but use markers that also label macrophages and microglia. How do the authors define microglial precursors and can they provide definitive data to show that these cells produce bonafide microglia. It is important to connect the precursor with the mature population. In other words, can the mpeg1.1:egfp cells in the retina be marked with bonafide microglia markers (such as apoe, lcp1, 4c4, tmem119, p2yr12). Are they clearing some sort of debris in the system that is expected of mature microglia, like for example in Figure 3D a", which clearly shows a cell with vacuoles. Varying morphology or phagocytosis may indicate an altered microglia, which is less consistent with a precursor population and more consistent with a microglia. Without this data, the cells can only be called myeloid precursors/progenitors.

Similarly, in the discussion, the authors note that "peripheral macrophages located.." but the data presented does not support the conclusion that macrophages vs microglia vs. microglia precursors can be distinguished. This is a critical point for the conclusions the authors present. By the authors definition, a mpeg+ cell outside the retina is a macrophage. But they also define a mpeg+ cell outside the retina as a microglial precursor. It is not clear how the authors can distinguish between these two cell-types. This is important in their quantifications as they manipulate different genes.

2. There are several controls that are important for the study. Please see the list below for revision.

Has zGem been established in zebrafish to label proliferating microglia. A positive control that indicates proliferating mpeg1.1+ cells can be detected with zGem is critical for this analysis.

Is there a positive control that can be quantified to indicate zGem is actually measuring dividing cells. For example, can mpeg+ cells in the periphery be scored?

On page 10, line 214 the authors make a bold claim that the ath5 morpholino data indicates microglia utilize the neurogenic region. However, it is possible that the ath5 MO also disrupts more than the neurogenic niche.

The experiments in the slbp1 background do not rule out the possibility that slbp1 could directly impact microglia precursor cells. Could the authors begin to rule this out by determining if microglia express slbp1?

Without detecting Il34 protein levels in overexpression experiment it is not clear whether the protein is even expressed or secreted. Do the authors have a positive control that the overexpression construct is in fact overexpressing Il34 and that it is secreted. Alternatively, do they have a positive control that the construct could induce a phenotype suggesting the construct is functional.

Morpholino experiments are used throughout the paper but clear controls for these morpholinos are not stated, cited or performed. Please refer to Stainier et al. 2017 paper to consider additional controls.

3. The authors claim that apoptosis is unlikely to be involved but further experiments are needed to definitely rule out that possibility.

Can the authors test the hypothesis of apoptosis by measuring levels of apoptosis during the time of colonization.

Can the authors demonstrate that the tp53 morpholino blocked apoptosis.

4. The main conclusion of the paper states that microglia enter the retina through a neurogenic niche. Through a series of mutants and drug treatments, the neurogeneic niche is altered. Unfortunately, all the manipulations to the neurogenic niche are global and thus it remains a possibility that microglia, microglia precursors and/or other tissues are also impacted by these manipulations. Although the mosaic experiments begin to address this concern, there is no measurement of the microglia from slbp1 mutant animals. Either additional experiments need to be included to directly rule out the possibility that microglia themselves are impacted or the conclusions of the paper need to be significantly restated.

The HDAC1/2 experiments are particularly difficult to interpret. HDAC1/2 inhibition also impairs microglial development and results in decreased cell number and altered morphology (Datta et al., 2018 Immunity). Prenatal ablation of Hdac1 and Hdac2 caused spontaneous microglia impairment, including blockage of proliferation and enhanced apoptosis while postnatal deletion was largely compatible with microglial viability and function. Cell specific manipulation of HDAC would need to be completed to ensure microglia are not impacted by HDAC dysfunction.

5. The mosaic experiments begin to address the cell-autonomy of the mutant backgrounds. This is a clever approach and I commend the authors for the brilliant experiment. However, it is not clear why certain comparisons were made to make the conclusions the authors made. Or it is possible that the mosaic experiment is confusing as written. Should not the comparison be between microglia associated with mutant cells vs wt cells in wt host retina or mutant cells vs wt cell in a mutant host retina. The wt to wt cell transplantation is more a control to determine if the transplants themselves cause retinal or microglia defects.

*Reviewer #3:*

Ranawat and Masai reports on characterizing the cellular mechanism by which microglia colonize the retina in zebrafish. This is an interesting and novel topic as there has not been a formal investigation of retinal microglia colonization in zebrafish or other vertebrate models, and would provide important insights into possible distinction, if any, between retinal versus brain microglia. The authors attempt to examine whether the same cellular mechanisms that mediate brain microglial colonization are the same for retinal microglia in zebrafish. However, many of the conclusions are coming from insufficient evidence or contain significant flaws, as discussed below. One of the major issues is that the authors attempt to make claims on several mechanistic features distinguishing retinal microglia from tectal brain microglia during colonization into the CNS, but the data either are missing important controls, or analysis is incomplete and remains quite equivocal.

1) The authors concluded that retinal microglial migration into retina depends on blood vessel network. Their primary argument was that tnnt2 morpholino-mediated blockage of blood flow disrupted blood vessel development, and thereby prevented retinal microglial colonization. However, this analysis still contains critical gaps (see a), and the alternate explanation that a lack of blood flow could prevent microglial colonization, which in turn disrupt blood vessel formation remains equally likely but not sufficiently addressed. They used pu.1 morpholino as a means for removal of myeloid cells (including microglia), but necessary control characterizations are lacking (see b).

a) Although colonization may begin earlier in the retina than in the tectum, analysis of whether blood flow/blood vessel is required should be verified at 3-4 dpf when colonization of both retinal and tectal microglia is abundant. In the same embryos when blood flow is blocked, both retina and brain microglia should be examined. This way the difference claimed by the authors that in fact retinal microglia are different from tectal microglia that they require blood flow/vessel could be clearly examined simultaneously to make a clear conclusion. Because of its small size at these stages, most studies of zebrafish microglia already image both retina and brain in whole mount, but the retinal microglia just have not been specifically analyzed, so there is no good reason for the authors not to examine both locations especially if they are making strong points about their differences. A priori, however, the gross level phenotypes of tectal microglia seem to align with retinal microglia based on papers already published about microglia mutants (including il34, irf8 etc).

b) The effectiveness of pu.1 mo to fully ablate macrophages was not demonstrated and in fact the panels shown in Figure 2 supp 1 indicate at best a partial ablation of myeloid cells in pu.1 morphants as many fluorescently red tagged macrophages were detectable. Since morpholino injections are transient, varying levels of pu.1 knockdown are expected, only individual embryos showing complete absence of macrophages should be used for analysis. Complementary experiments should be done using macrophage-lacking mutants such as irf8 mutants (this will address any caveat from transient/mosaic knockdown by morpholino) to confirm a lack of a macrophage role in retinal vessel development. Appropriate validation is critical for making correct interpretations, especially in light of the negative data and also previous work in mice implicating microglia for retinal vessel development.

2) Data as shown in Figure 1-supp3 lead to questions of whether "dividing" microglial cells were reliably quantified. The lateral view is at low resolution in a thick section of the retina-the example does not show any convincing example of a single microglial cell that is also BrdU^+^. Please analyze at higher cellular resolution and show a magnified example of a proliferative microglial cell. The concern is that the counting is over-represented than the actual number dividing which looks like zero in the A panels example. There seems to be far fewer if any dividing microglial cell.

3) The authors claim a second feature relating to not requiring signals from apoptotic neurons that sets apart retinal from tectal microglia in zebrafish, but the evidence is weak. Based on studies of others, microglia have been shown to colonize the tectum in part by signals coming from apoptotic neurons in the brain in zebrafish. This current study only transiently inhibits global cell death using an antisense morpholino against tp53. At the minimum, the authors need to show that (1) this tp53 MO injection does indeed eliminate retinal/brain apoptosis during development, (2) that at later stages 3-5 dpf using tp53 morpholino (assuming they confirm its efficacy in inhibiting neuronal apoptosis), if their conclusion is correct, the results should show normal presence of retinal microglia but an absence of tectal microglia, and (3) slbp1 mutants do not have less apoptotic neurons to explain for a reduced microglial colonization in the retina.

4) Histone deacetylase 1 (HDAC1) is expected to function in a variety of cell type. The use of its chemical inhibitor TSA broadly, and the global hdac1 mutant raise question as to whether hdac1 itself has a function in macrophages to affect their ability to migrate or differentiate into microglia? Figure 4-supp3 shows some punctate macrophage reporter patterns. The authors need to address this.

5) Whether a morpholino has been published previously or not, efficacy for all morpholinos should be verified either at least phenotypically (if previously published), or by both molecular and phenotypic means (if new and yet unpublished) to substantiate that the reagent works as intended in the researcher's own laboratory operations. This includes the cxcl12a morpholino.

6) The interpretation that il34/csf1r pathway is not involved in retinal microglial colonization as it is for tectal brain microglia in zebrafish is surprising, because recent analyses of microglia in zebrafish il34 mutants indicate a significant reduction (based on data from these work there is reduction also in retina, but the retinal population was not formally analyzed) (Kuil et al., 2019 DMM; Wu et al., 2018 Dev Cell). The evidence on which the authors have made this conclusion is rather weak and possibly flawed. First of all, gene expressions on whole body for il34 distinguishing WT vs slbp1 mutant and that csf1r not being detected in a single cell analysis of microglial precursors cannot form a strong argument that this pathway is not involved as csf1r may just be lowly expressed and global expression may not be informative. The authors must analyze the il34 mutants to examine whether retinal microglia colonize or not in parallel with the tectal microglia. The only functional experiment they use is an rx1:il34 overexpression in the eye which was not sufficient to increase macrophage colonization-however the caveat is that this overexpression is mosaic and whether functional il34 proteins are getting produced is not known.

7) Finally, more data points are required to formulate a reliable conclusion that microglia have a preference for differentiated retinal neurons and less with progenitors using the chimeric host-donor transplantation. There is a big differential in number of neural columns transplanted in mutants (more donor cells 3-7 columns in mutant hosts, than the 2-4 columns in control wt host group), which can misleadingly show an increased association of microglia with mature columns just because there are more columns in the slbp1 mutants. More Ns to establish equivalent range of donor transplantation (especially larger numbers in the wt group), and also the reciprocal test of putting donor mutant columns into WT host, will provide the needed comprehensive test of this concept using chimeras.

8) The use of slbp1 mutants being only interpreted as a neurogenesis mutant can be problematic as this gene slbp1 (stem-loop binding protein binds to 3'end histone mRNAs) regulates degradation of histone proteins and replication-dependent synthesis and can have broad effects similar to the issue with hdac1 mutants. Complementary method to create neuron-specific disruption of neurogenesis would be ideal to substantiate the concept that the differentiation status of retinal neurons strongly affect colonization of microglia in the retina.

[Editors’ note: further revisions were suggested prior to acceptance, as described below.]

Thank you for resubmitting your work entitled "Mechanisms underlying microglial colonization of developing neural retina in zebrafish" for further consideration by *eLife*. Your revised article has been reviewed by 3 peer reviewers and the evaluation has been overseen by Richard White (Senior Editor) and a Reviewing Editor.

Overall, the authors have done a thorough and comprehensive job of addressing the points raised by the initial peer review and added additional controls and significant new data. This has particularly strengthened the conclusion that microglia colonization of retina is linked to neurogenesis, which is the central new finding of this study. However, there are a few remaining (minor) concerns that need to be addressed as summarized below by reviewers 1 and 3. In addition, there are few areas in the manuscript that overstate the conclusions based on the data reported in the manuscript. Please consider modifying the following conclusions it is not feasible to address experimentally in a timely manner.

– The origin and characterization of the macrophage population(s) in the retina. For example, please state clearly their definition of microglia early in the results (In this manuscript, we define microglia as…").

– The spatiotemporal role and relationship of IL34, microglia colonization and neurogenesis as raised by reviewer 1. These concerns can be addressed by adding experiments or modifying the current conclusions.

*Reviewer #1:*

Please consider the following suggestions for the paper.

– The authors should use other markers to verify that the cell populations of interest are microglial precursors. Using only macrophage markers may not encapsulate the heterogeneity of embryonic microglia in the zebrafish. Can the authors analyze other markers for microglia to demonstrate their identity? While they cite papers referencing apoeb, the data in the manuscript demonstrate microglia in different CNS regions have distinct properties. There are several reagents like antibodies against 4C4, or L-plastin or in situ hybridization of microglia genes like apoeb, or p2ry12 that could be used, as done in other zebrafish microglia manuscripts.

– The systemic inhibition of circulation is a concern because the manipulation is a global perturbation that impacts heart contraction. The data supporting the reliance of migrating microglial precursors on vasculature would be more convincing if the authors specifically inhibited the vasculature that they propose is utilized for microglia colonization. This concern is heightened by the observation that the animals in the tnnt2aMO look very unhealthy.

– There are a few areas in the manuscript that overstate the conclusions based on the data reported in the manuscript. Please consider modifying the following conclusions in the manuscript:

– The authors summarize their findings with a schematic in figure 6. The model states a stepwise process to microglia colonization of the retina, but the data reported in the manuscript support a more continuous process of microglia colonization. Further, the first step in the model is IL34-mediated migration to the brain, but the data reported in the manuscript only shows a reduction of microglia in the retina without follow up experiments to address when in the process microglia are disrupted in IL34 mutant animals. Please consider revising the model so that it is more consistent with the conclusions in the paper.

– The comment "it is very likely that csf1r-il34 signaling promotes…" overstates the conclusion one can draw from the reported data. It would be more accurate to state that it is involved in colonization. Otherwise, the authors should assess the step wise process of colonization when IL34 is perturbed.

– The authors utilize the NICD manipulation to test whether neurogenesis is required for microglia colonization of the retina. This is a clever approach to altering neurogenesis. With this experiment, it is possible that the Notch pathway is directly impacting microglia colonization. Please consider modifying the sentence line 379-380 "Thus microglial precursors are less attracted…in which neurogenesis was arrested." to add a disclaimer that the direct role of Notch signaling cannot be ruled out.

*Reviewer #2:*

In this study Ranawat and Masai investigate the routes by which microglia colonize the zebrafish retina. The authors show that, after migrating to the eye and entering through hyaloid vessels, microglial precursors are recruited into the retina by neural progenitors exiting the cell cycle. Using three different genetic models of delayed retinal neurogenesis (ath5 morphants, slbp1 mutants, and overexpression of notch intracellualar domain), they show a delay in microglial entry into the retina when neurogenesis is suppressed. Further, in genetic chimeric retinas, microglia preferentially associate with neurogenic radial clones rather than clones of neurogenically-impaired cells.

Overall, this study is very interesting and contributes several new findings that will be of broad interest. These include: (1) the observation that mechanisms for microglial colonization differ between brain and retina; and (2) a holistic overview of multiple cellular and molecular mechanisms that guide microglial precursors from the site of their birth into the neural retina. A minor weakness is that the authors have not identified even a hint of the molecular cues that might explain the ability of newborn neurons to recruit microglial entry into the retina. Nevertheless, given the other strengths of this paper and its contributions, I do not think this weakness is a particularly severe one.

I have no concerns with the data or their interpretation. It is my impression that the authors have done a thorough and comprehensive job of addressing the points raised by the initial peer review. The authors are to be commended on a compelling and clearly-written study.

*Reviewer #3:*

In this study, the authors investigate microglia colonization of the retina in zebrafish. Some of their observations confirm work from previous studies. They first carefully show that there is almost no proliferation of microglial precursors, which was previously shown in mouse retina (Santos et al., 2008) but not in zebrafish. They make the sound conclusion that microglia precursors migrate in to colonize the retina. They clearly show that neuronal apoptosis is not the major cue for microglial precursor colonization of the retina as it is for tectum, although this was previously documented in zebrafish by Wu et al., 2018 (which they gloss over in the manuscript). They also show that Csf1r-il34 signaling is important for microglia colonization of retina, although this was also shown by Wu et al. 2018 (again glossed over). Due to almost complete failure of microglia precursors to populate they eye in IL34 mutants, they argue that Csf1r-il34 signaling promotes microglial precursor movement from yolk to the optic cup upstream of the blood vessel-mediated guidance mechanism. However, it remains unclear which cell populations express IL34 developmentally to attract microglia precursors to the eye and at what stages this occurs. So their observations regarding the role of Csf1r-il34 signaling do not advance much beyond what is shown by Wu et al., 2018.

They also make some new observations. They convincingly show that entry of microglial precursors into the optic cup through the choroid fissure depends on ocular blood vessels. Most interestingly, they show using three different manipulations that delaying retinal neurogenesis reduces microglial precursor colonization of the retina. They further show using elegant transplant studies that microglia precursors have greater affinity for differentiating neurons than for retinal progenitor cells. These experiments are carefully done and convincing. The mechanisms underlying this process are unclear since RGC differentiation begins at 27hpf, but microglia precursors don't enter until after 42hpf. They rule out IL34 as the signal since it is not altered in their slbp1 mutants (which show delayed neurogenesis and delayed microglia colonization). So the main new conclusion is that microglia colonization of retina depends upon ocular blood vessels and is linked to neurogenesis. These findings are interesting and important.

The authors have thoughtfully responded to prior comments from reviewers and added additional controls and significant new data. This has particularly strengthened the conclusion that microglia colonization of retina is linked to neurogenesis, which is the central new finding of this study. However, the mechanisms underlying this process remain undefined.

---

## [Author Response]

[Editors’ note: the authors resubmitted a revised version of the paper for consideration. What follows is the authors’ response to the first round of review.]

Reviewer #2:1. The authors use the term microglial precursors often but use markers that also label macrophages and microglia. How do the authors define microglial precursors and can they provide definitive data to show that these cells produce bonafide microglia. It is important to connect the precursor with the mature population. In other words, can the mpeg1.1:egfp cells in the retina be marked with bonafide microglia markers (such as apoe, lcp1, 4c4, tmem119, p2yr12). Are they clearing some sort of debris in the system that is expected of mature microglia, like for example in Figure 3D a", which clearly shows a cell with vacuoles. Varying morphology or phagocytosis may indicate an altered microglia, which is less consistent with a precursor population and more consistent with a microglia. Without this data, the cells can only be called myeloid precursors/progenitors.Similarly, in the discussion, the authors note that "peripheral macrophages located.." but the data presented does not support the conclusion that macrophages vs microglia vs. microglia precursors can be distinguished. This is a critical point for the conclusions the authors present. By the authors definition, a mpeg+ cell outside the retina is a macrophage. But they also define a mpeg+ cell outside the retina as a microglial precursor. It is not clear how the authors can distinguish between these two cell-types. This is important in their quantifications as they manipulate different genes.

In zebrafish, early macrophages are generated from myeloid cells originating in the rostral blood island (RBI) around 11 hpf and colonize the brain and retina by 55 hpf (Xu et al., 2015). At around 60 hpf, these brain and retina-colonized macrophages undergo a phenotypic transition, which indicates expression of mature microglial markers such as apolipoprotein E (apoE) and phagocytic behavior toward dead cells (Herbomel et al., 2001). Importantly, early macrophages outside the brain never express apoE (Herbomel et al., 2001), suggesting that only brain and retina-resident macrophages give rise to microglia. Thus, early macrophages colonizing the brain and retina by 60 hpf are generally accepted as microglial precursors in zebrafish. Herein, we refer to mpeg1.1:EGFP^+^ and mfap4:tdTMTCAAX^+^ cells in the brain and retina at 60 hpf as microglial precursors. It is also appropriate to refer to mpeg1.1:EGFP^+^ and mfap4:tdTMT-CAAX^+^ cells outside the retina and brain as macrophages, although these macrophages may have the potential to differentiate into microglial precursors after they enter the brain and retina. We have added this definition of microglial precursors, microglia, and macrophages to the first paragraph of the Results section (page 6, line 165-173).

2. There are several controls that are important for the study. Please see the list below for revision.Has zGem been established in zebrafish to label proliferating microglia. A positive control that indicates proliferating mpeg1.1+ cells can be detected with zGem is critical for this analysis.Is there a positive control that can be quantified to indicate zGem is actually measuring dividing cells. For example, can mpeg+ cells in the periphery be scored?

First, we confirmed the presence of mCherry-zGem and mpeg1.1:EGFP double-positive cells in peripheral tissues and found that more than 60 % of mpeg1.1:EGFP-positive cells expressed mCherryzGem, suggesting that this *Tg[EF1*a*:mCherry-zGem]* system functions as an indicator of cell cycle phases in early zebrafish macrophages. Furthermore, to confirm that a majority of ocular microglial precursors do not undergo S phase, by labeling of retinal sections with anti-BrdU antibody, we confirmed that more than 80% of mpeg1.1:EGFP^+^ cells did not incorporate BrdU at 48 hpf, suggesting that microglial precursor colonization of the retina depends mostly on cell migration from outside the optic cup, rather than on precursor cell proliferation. We have added these results to Figure 1—figure supplement 3 and 4 as new data.

On page 10, line 214 the authors make a bold claim that the ath5 morpholino data indicates microglia utilize the neurogenic region. However, it is possible that the ath5 MO also disrupts more than the neurogenic niche.

The aim of the ath5 morpholino experiment was to investigate the possibility that infiltration of microglial precursors into the neural retina depends on RGC differentiation or RGC-mediated circuit formation. In the original and revised manuscripts, we showed that microglial precursors infiltrate the neural retina preferentially through the neurogenic region, and that microglial precursor infiltration of the neural retina depends on retinal neurogenesis, suggesting that retinal neurogenesis functions as a gateway. However, since a blockade of retinal neurogenesis delays retinal cell differentiation and subsequent neural circuit formation, it is possible that differentiation of the first-born retinal cell-type, RGCs, is required for infiltration of microglial precursors into the neural retina.

We previously showed that ath5 starts to be expressed in G2 phase of retinal progenitor cells just prior to their final neurogenic cell division, and then ath5 expression is inherited by their daughter cells (Poggi et al., 2005, J. Cell Biol., 171, 991-999.; Yamaguchi et al., 2010, Mech Dev 127, 247-264.). Although ath5 is thought to be expressed in all retinal neurogenic lineages, only RGCs fail to differentiate in the zebrafish *ath5* mutant, *lakritz* (Kay et al., 2001). We also confirmed that RGC differentiation is compromised in *ath5* morphant retinas (Figure 4—figure supplement 8AB). Thus, *ath5* morphants provide a good platform for us to investigate whether RGC differentiation is required for microglial precursor infiltration into the neural retina. We found that ocular microglial precursors were significantly reduced in number in *ath5* morphant retinas. So, defects in RGC differentiation affect microglial precursor infiltration of the neural retina. We currently consider it likely that RGC differentiation or RGC-mediated IPL formation is required for proper timing of microglial precursor infiltration of the neural retina. We have revised the description on *ath5* morphant data (page 12, line 461- page 13, line 523) to clarify this possibility.

The experiments in the slbp1 background do not rule out the possibility that slbp1 could directly impact microglia precursor cells. Could the authors begin to rule this out by determining if microglia express slbp1?

We agree with reviewer 2 that it is possible that the *slbp1* mutation may impact behavior of microglial precursor cells. Slbp1 binds to the stem loop structure of 3’-untranslated region of histone mRNA and promotes translation of histone mRNA. Thus, Slbp1 is a key enzyme of histone genesis. In *slbp1* mutants, histone mRNAs are abnormally polyadenylated, and the conventional translation mechanism for poly-adenylated mRNAs rescues histone mRNA translation. However, this conventional translation mechanism does not fully ensure histone genesis in highly proliferating cells such as retinal progenitor cells, so cell-cycle progression becomes slow in *slbp1* mutant retinas (Imai et al., 2014 Dev Biol 394, 94-109.). On the other hand, we did not observe any defects in the number of mfap4^+^ cells in *slbp1* mutant embryos, except the reduction of ocular microglial precursors (see Figure 4CD). Since a majority of microglial precursors do not undergo S phase, we think that the conventional translation mechanism may be enough to support microglial precursor proliferation in *slbp1* mutants.

To confirm our conclusion on neurogenesis-mediated infiltration of microglial precursors into the retina, we have added the data on NICD overexpression in zebrafish retina, which reduced ocular microglial precursors. Please see our response to reviewer 2’s comment 4.

Without detecting Il34 protein levels in overexpression experiment it is not clear whether the protein is even expressed or secreted. Do the authors have a positive control that the overexpression construct is in fact overexpressing Il34 and that it is secreted. Alternatively, do they have a positive control that the construct could induce a phenotype suggesting the construct is functional.

We agree with reviewer 2’s concern regarding the data about IL34 overexpression. Unfortunately, we do not have a good antibody that immunohistochemically recognizes zebrafish IL34. Furthermore, *il34* mRNA is very difficult to detect by in situ hybridization. To evaluate whether IL34 is involved in microglial precursor infiltration of the retina, we gave up this overexpression experiment, but we investigated the number of ocular microglial precursors in zebrafish *il34* mutants. We found that the number of ocular microglial precursors was severely reduced. Very few microglial precursors (from 0 to 2 cells) are associated with blood vessels in *il34* mutant eyes at 48 hpf. These data suggest that IL34 is required for microglial precursor colonization of the retina in zebrafish. Since almost no microglial precursors enter the optic cup, it is very likely that IL34 promotes guidance of microglial precursors from the yolk to the retina, upstream of blood vessel-mediated entry of microglial precursors into the optic cup and neurogenesis-mediated infiltration of microglial precursors into the neural retina. We have added these new data in new Figure 6AB and have revised our conclusion as follows: There are three steps for microglial colonization of the retina: In the first, IL34 attracts microglial precursors and promotes their migration into the brain and eye. In the second step, blood vessels guide microglial precursors to enter the optic cup. In the third step, retinal neurogenesis enables microglial precursors to infiltrate the neural retina. This conclusive summary is also provided in new Figure 6C.

Morpholino experiments are used throughout the paper but clear controls for these morpholinos are not stated, cited or performed. Please refer to Stainier et al. 2017 paper to consider additional controls.

We apologize for having failed to include the description of standard MO in the Materials and methods section. We have added the information on sequence and concentration of standard MO.

In this study, we used 5 morpholinos: tnnt2a MO, p53 MO, cxcl12a MO, pu.1 MO, and ath5 MO. All of these MOs, except ath5 MO, were previously reported. However, to evaluate whether each MO concentration is enough to knock down target gene translation, we confirmed that each MO induced expected phenotypic defects, as shown below. We have added these data to the supplementary figures below. tnnt2a MO: thinner blood vessels shown in Figure 2C and Figure 2—figure supplement 2A.

p53 MO: Reduction of apoptosis shown in Figure 3—figure supplement 2. cxcl12a MO: Misrouted trajectory of retinal axons shown in Figure 4—figure supplement 5AB. pu.1 MO: No microglial precursors shown in Figure 2—figure supplement 3AB. ath5 MO: No RGC differentiation shown in Figure 4—figure supplement 8.

3. The authors claim that apoptosis is unlikely to be involved but further experiments are needed to definitely rule out that possibility.Can the authors test the hypothesis of apoptosis by measuring levels of apoptosis during the time of colonization.Can the authors demonstrate that the tp53 morpholino blocked apoptosis.

We confirmed that p53 MO significantly suppressed apoptosis in zebrafish retina at 24 and 36 hpf. At 48 hpf, the apoptotic level was lower than that of 24 and 36 hpf in wild-type retina, so there was no statistical difference in apoptotic cell number between Standard MO injected retinas and p53 MO injected retinas. We have added these data to Figure 3—figure supplement 2AB.

In addition, we investigated colonization of microglial precursors in retina at 48 hpf and optic tectum at 96 hpf in zebrafish, and confirmed that microglial precursor colonization of the retina was not changed in *p53* morphants; however, microglial colonization of the optic tectum was affected in *p53* morphants. These data suggest that colonization of the optic tectum depends on apoptosis, which is consistent with previous reports. On the other hand, colonization of the retina is independent of apoptosis. We have added these data to Figure 3—figure supplement 3.

4. The main conclusion of the paper states that microglia enter the retina through a neurogenic niche. Through a series of mutants and drug treatments, the neurogeneic niche is altered. Unfortunately, all the manipulations to the neurogenic niche are global and thus it remains a possibility that microglia, microglia precursors and/or other tissues are also impacted by these manipulations. Although the mosaic experiments begin to address this concern, there is no measurement of the microglia from slbp1 mutant animals. Either additional experiments need to be included to directly rule out the possibility that microglia themselves are impacted or the conclusions of the paper need to be significantly restated.

We previously reported that Notch1 intracellular domain (NICD) suppresses retinal neurogenesis in zebrafish (Yamaguchi et al., 2005). To inhibit retinal neurogenesis more directly without perturbation of microglial precursor functions, we overexpressed NICD in retinal cells and examined ocular microglial colonization of the retina. First we confirmed that overexpression of NICD significantly suppresses retinal neurogenesis in zebrafish by injecting a DNA expression construct encoding UAS: myc-tagged NICD into *Tg[hs:gal4; ath5:EGFP]* double transgenic embryos (Figure 4—figure supplement 7).

Next, we established a zebrafish transgenic line, *Tg[rx1:gal4-VP16],* in which gal4-VP16 is expressed under control of the *rx1* promoter, which drives mRNA expression in retinal progenitor cells. Then, we injected a mixture of two DNA expression constructs encoding UAS:myc-tagged NICD and UAS:EGFP into *Tg[rx1:gal4-VP16; mfap4:tdTMT-CAAX]* embryos. We selected embryos in which EGFP was expressed in most retinal cells at 24 hpf, and used them to investigate the number of ocular microglial precursors. We found that the number of ocular microglial precursors was significantly decreased in retinas overexpressing NICD, compared with retinas expressing only EGFP. Since the *rx1* promoter does not drive NICD expression in microglial precursors, these data suggest microglial colonization of the neural retina depends on retinal neurogenesis. We have added these new data to Figure 4EF.

In addition, we performed more sparse mosaic expression of NICD in wild-type retinas by injection of a DNA construct encoding UAS:EGFP or a mixture of UAS:myc-NICD and UAS:EGFP into *Tg[hsp:gal4: mfap4:tdTMT-CAAX]* transgenic embryos. A fraction of microglial precursors associated with EGFP-positive columns was lower in UAS:myc-NICD+UAS:EGFP expression than in only UAS:EGFP expression. Trapping efficiency of microglial precursors in each EGFP-positive column was also lower in UAS:myc-NICD+UAS:EGFP expression than in only UAS:EGFP expression. We have added these data to Figure 5EFGH.

Taken together, these data strongly suggest that retinal neurogenesis drives microglial precursors to infiltrate the retina.

The HDAC1/2 experiments are particularly difficult to interpret. HDAC1/2 inhibition also impairs microglial development and results in decreased cell number and altered morphology (Datta et al., 2018 Immunity). Prenatal ablation of Hdac1 and Hdac2 caused spontaneous microglia impairment, including blockage of proliferation and enhanced apoptosis while postnatal deletion was largely compatible with microglial viability and function. Cell specific manipulation of HDAC would need to be completed to ensure microglia are not impacted by HDAC dysfunction.

We agree with reviewer 2’s concern regarding HDAC1 mutant data. We have deleted these results from our manuscript.

5. The mosaic experiments begin to address the cell-autonomy of the mutant backgrounds. This is a clever approach and I commend the authors for the brilliant experiment. However, it is not clear why certain comparisons were made to make the conclusions the authors made. Or it is possible that the mosaic experiment is confusing as written. Should not the comparison be between microglia associated with mutant cells vs wt cells in wt host retina or mutant cells vs wt cell in a mutant host retina. The wt to wt cell transplantation is more a control to determine if the transplants themselves cause retinal or microglia defects.

We previously showed that in chimeric retinas consisting of wild-type and *slbp* mutant cells, *slbp1* mutant retinal columns showed a delay of neurogenesis and maintained a proliferative state, whereas wild-type retinal columns show a normal temporal program of retinal neurogenesis and neuronal differentiation (Figure 3 and 6 of “Imai et al., (2014) *Dev. Biol.*, 394, 94-109.”). Thus, we investigated the extent to which wild-type donor retinal cells attract microglial precursors in *slbp* mutant retinas by competing with *slbp* mutant host retinal cells. In this case, a simple control is transplantation of wildtype donor cells into wild-type host retinas. It is ideal to investigate the rate of association of microglial precursors with donor retinal columns under similar conditions with the total number of host microglial precursors per eye and the total number of transplanted donor columns per eye.

In the revised manuscript, we presented all this information plus the number of microglial precursors associated with transplanted retinal columns for each sample: 5 samples of wt-> wt and 3 samples of wt -> mut in Figure 5—figure supplement 1. The total number of host microglial precursors at 48 hpf was n=19.6 for wt-> wt and n=7.7 for wt-> mut, which is consistent with data shown in Figure 4AB. The total number of transplanted donor columns is n=5.6 for wt-> wt and n=4.7 for wt-> mut. Thus, we believe that comparison of the fraction of microglial precursors associated with donor retinal columns between wt -> wt and wt -> mut was done appropriately to evaluate trapping efficiency for donor retinal cells to host microglial precursors.

Furthermore, we added another chimeric retina experiment by introducing retinal columns expressing NICD in wild-type retinas. In this case, we introduced a small number of (NICD-expressing) proliferative columns in wild-type neurogenic retinas, which is contrast to the transplantation experiment from wild-type donor cells into *slbp* mutant host retinas. We obtained consistent data, which support the idea that retinal neurogenic columns are more attractive to microglial precursors than proliferative progenitor columns. These data have been added to Figure 5E-H.

Reviewer #3:Ranawat and Masai reports on characterizing the cellular mechanism by which microglia colonize the retina in zebrafish. This is an interesting and novel topic as there has not been a formal investigation of retinal microglia colonization in zebrafish or other vertebrate models, and would provide important insights into possible distinction, if any, between retinal versus brain microglia. The authors attempt to examine whether the same cellular mechanisms that mediate brain microglial colonization are the same for retinal microglia in zebrafish. However, many of the conclusions are coming from insufficient evidence or contain significant flaws, as discussed below. One of the major issues is that the authors attempt to make claims on several mechanistic features distinguishing retinal microglia from tectal brain microglia during colonization into the CNS, but the data either are missing important controls, or analysis is incomplete and remains quite equivocal.1) The authors concluded that retinal microglial migration into retina depends on blood vessel network. Their primary argument was that tnnt2 morpholino-mediated blockage of blood flow disrupted blood vessel development, and thereby prevented retinal microglial colonization. However, this analysis still contains critical gaps (see a), and the alternate explanation that a lack of blood flow could prevent microglial colonization, which in turn disrupt blood vessel formation remains equally likely but not sufficiently addressed. They used pu.1 morpholino as a means for removal of myeloid cells (including microglia), but necessary control characterizations are lacking (see b).a) Although colonization may begin earlier in the retina than in the tectum, analysis of whether blood flow/blood vessel is required should be verified at 3-4 dpf when colonization of both retinal and tectal microglia is abundant. In the same embryos when blood flow is blocked, both retina and brain microglia should be examined. This way the difference claimed by the authors that in fact retinal microglia are different from tectal microglia that they require blood flow/vessel could be clearly examined simultaneously to make a clear conclusion. Because of its small size at these stages, most studies of zebrafish microglia already image both retina and brain in whole mount, but the retinal microglia just have not been specifically analyzed, so there is no good reason for the authors not to examine both locations especially if they are making strong points about their differences. A priori, however, the gross level phenotypes of tectal microglia seem to align with retinal microglia based on papers already published about microglia mutants (including il34, irf8 etc).

In accordance with reviewer 3’ suggestion, we examined microglial colonization of the optic tectum at 48hpf and 72 hpf. The number of mfap4^+^ cells in the optic tectum was ~5 at 48 hpf and increased to ~30 at 72 hpf in standard MO-injected control embryos, indicating progressive increase of tectal microglia in wild-type embryos during the period from 48 hpf to 72 hpf. On the other hand, the number of mfap4^+^ cells in the optic tectum was ~15 and significantly higher in tnnt2a-MO-injected embryos than in standard MO-injected control embryos at 48 hpf. In addition, tectal mfap4^+^ cells in *tnnt2a* morphants at this stage (48 hpf) displayed a round shape, which is reminiscent of a phagocytic activation state. We suspect that apoptosis may be increased in the optic tectum of *tnnt2a* morphants, attracting more microglia into the optic tectum at 48 hpf. However, the number of mfap4^+^ cells in the optic tectum was not drastically increased in *tnnt2a* morphants from 48 hpf to 72 hpf, so the number of tectal mfap4^+^ cells is not significantly different from that of standard MO-injected control embryos at 72 hpf, as reported previously. Thus, the number of mfap4^+^ cells in the optic tectum is eventually similar in standard MO-injected control embryos and tnnt2a-MO-injected embryos at 72 hpf. Although our tnnt2a-MO data are consistent with the previous report, we think that the situation may be more complex and differs from the previous report, which concluded that microglial colonization of the optic tectum does not depend on blood vessel function in zebrafish. We have added these observations to Figure 2 —figure supplement 2. These data support our conclusion that microglial precursor colonization of the retina depends on blood vessels at 48 hpf, in contrast to the optic tectum at 3 dpf.

b) The effectiveness of pu.1 mo to fully ablate macrophages was not demonstrated and in fact the panels shown in Figure 2 supp 1 indicate at best a partial ablation of myeloid cells in pu.1 morphants as many fluorescently red tagged macrophages were detectable. Since morpholino injections are transient, varying levels of pu.1 knockdown are expected, only individual embryos showing complete absence of macrophages should be used for analysis. Complementary experiments should be done using macrophage-lacking mutants such as irf8 mutants (this will address any caveat from transient/mosaic knockdown by morpholino) to confirm a lack of a macrophage role in retinal vessel development. Appropriate validation is critical for making correct interpretations, especially in light of the negative data and also previous work in mice implicating microglia for retinal vessel development.

We apologize that our preparation of Figure 2—figure supplement 1 in the original manuscript was not good enough. Patchy dotted magenta signals surrounding the optic cup in pu.1 morphants were caused by noise caused by reflection of iridophore pigments, which start to appear after 48 hpf. Furthermore, we did not clearly indicate the ocular microglial precursors in standard MO-injected retinas, because mfap4 fluorescent signals in the current 3D images contain both ocular microglial precursors as well as peripheral macrophages located in mesenchymal area between the optic cup and the diencephalon. Thus, we re-examined 3D scanning of pu.1 morphants and extracted only ocular microglial precursors using the Imaris software surface rendering tool, and we confirmed that there are no ocular microglial precursors in pu.1 morphant embryos. We have replaced the previous figure with new scanning image data in new Figure 2—figure supplement 3AB. Furthermore, in accordance with reviewer 3’s suggestion, we examined blood vessel formation in *irf8* mutant retinas, and confirmed that ocular blood vessel formation is normal in i*rf8* mutants. We have added these data to new Figure 2—figure supplement 3CD.

2) Data as shown in Figure 1-supp3 lead to questions of whether "dividing" microglial cells were reliably quantified. The lateral view is at low resolution in a thick section of the retina-the example does not show any convincing example of a single microglial cell that is also BrdU^+^. Please analyze at higher cellular resolution and show a magnified example of a proliferative microglial cell. The concern is that the counting is over-represented than the actual number dividing which looks like zero in the A panels example. There seems to be far fewer if any dividing microglial cell.

In accordance with the reviewer’s suggestion, we conducted BrdU labeling of retinal sections of mpeg1.1:EGFP transgenic embryos, and confirmed that the fraction of BrdU^+^; mpeg1.1:EGFP+ cells among total mpeg1.1:EGFP+ cells is less than 20% on average, suggesting that more than 80% of microglial precursors do not undergo S phase. We have revised the BrdU data and it is shown in Figure 1—figure supplement 4BCD.

3) The authors claim a second feature relating to not requiring signals from apoptotic neurons that sets apart retinal from tectal microglia in zebrafish, but the evidence is weak. Based on studies of others, microglia have been shown to colonize the tectum in part by signals coming from apoptotic neurons in the brain in zebrafish. This current study only transiently inhibits global cell death using an antisense morpholino against tp53. At the minimum, the authors need to show that (1) this tp53 MO injection does indeed eliminate retinal/brain apoptosis during development, (2) that at later stages 3-5 dpf using tp53 morpholino (assuming they confirm its efficacy in inhibiting neuronal apoptosis), if their conclusion is correct, the results should show normal presence of retinal microglia but an absence of tectal microglia, and (3) slbp1 mutants do not have less apoptotic neurons to explain for a reduced microglial colonization in the retina.

We appreciate this suggestion. A similar concern was raised by reviewer 2, in suggestion (3).

In accordance with reviewer 3’s suggestion, we examined efficiency of p53 MO to inhibit apoptosis in the retina at 24, 36, and 48 hpf. We confirmed that p53 MO significantly suppressed apoptosis in zebrafish retina at 24 and 36 hpf. At 48 hpf, the apoptotic level was lower than that at 24 and 36 hpf in wild-type retina, so there was no statistical difference in apoptotic cell number between Standard MO-injected retinas and p53 MO-injected retinas. We have added these data in Figure 3—figure supplement 2AB.

Furthermore, we investigated colonization of microglial precursors into tectum at 96 hpf in zebrafish. Although microglial precursor colonization of the retina was not changed in *p53* morphants, microglial colonization of the optic tectum was affected in *p53* morphants. These data suggest that colonization of the optic tectum depends on apoptosis, which is consistent with previous reports. On the other hand, colonization of the retina is independent of apoptosis. We have added these data to Figure 3—figure supplement 3.

We are grateful for this suggestion to investigate apoptosis in *slbp* mutants. We employed TUNEL in wild-type and *slbp* mutant retinas at 48 hpf. We found that apoptotic cells were significantly increased in *slbp* mutant retinas. These data exclude the possibility that decreased retinal apoptosis affects microglial precursor colonization of the retina in *slbp* mutants, and again confirmed that neuronal apoptosis is not a major cue for microglial precursor colonization of the retina in zebrafish. We have added these data to Figure 4—figure supplement 4.

From these observations, we consider it likely that retinal neurogenesis is delayed in *slbp* mutants -> microglial precursor infiltration is compromised -> naturally occurring apoptotic cells fail to be eliminated because of the reduced number of microglial precursors in *slbp* mutant retinas; however, interestingly, these increased dead cells do not promote microglial precursor infiltration into *slbp* mutant retinas, suggesting that neurogenesis primarily opens the gate through which microglial precursors enter the retina. Since retinal neurogenesis normally occurs from 24 to 48 hpf in zebrafish, microglial precursors could not be attracted by apoptosis without the infiltration path opened by retinal neurogenesis before 48 hpf. This interpretation could explain why microglial precursors approaching towards the retina before 48 hpf are insensitive to apoptosis. We have added this idea to the discussion.

4) Histone deacetylase 1 (HDAC1) is expected to function in a variety of cell type. The use of its chemical inhibitor TSA broadly, and the global hdac1 mutant raise question as to whether hdac1 itself has a function in macrophages to affect their ability to migrate or differentiate into microglia? Figure 4-supp3 shows some punctate macrophage reporter patterns. The authors need to address this.

A similar concern was raised by reviewer 2, in suggestion (4).

We agree with reviewer 3, so we have deleted the HDAC data from our manuscript.

5) Whether a morpholino has been published previously or not, efficacy for all morpholinos should be verified either at least phenotypically (if previously published), or by both molecular and phenotypic means (if new and yet unpublished) to substantiate that the reagent works as intended in the researcher's own laboratory operations. This includes the cxcl12a morpholino.

In this study, we used 5 morpholinos: tnnt2a MO, p53 MO, cxcl12a MO, pu.1 MO, and ath5 MO. All MOs except ath5 have been previously reported. However, to evaluate whether each MO concentration is sufficient to knock down target gene translation, we confirmed that each MO induced the expected phenotypic defects. We have added these data to the supplementary figures below.

tnnt2a MO: thinner blood vessels shown in Figure 2C and Figure 2—figure supplement 2A.

p53 MO: Reduction of apoptosis shown in Figure 3—figure supplement 2. cxcl12a MO: Misrouted trajectory of retinal axons shown in Figure 4—figure supplement 5AB. pu.1 MO: No microglial precursors shown in Figure 2—figure supplement 3AB. ath5 MO: No RGC differentiation shown in Figure 4—figure supplement 8.

6) The interpretation that il34/csf1r pathway is not involved in retinal microglial colonization as it is for tectal brain microglia in zebrafish is surprising, because recent analyses of microglia in zebrafish il34 mutants indicate a significant reduction (based on data from these work there is reduction also in retina, but the retinal population was not formally analyzed) (Kuil et al., 2019 DMM; Wu et al., 2018 Dev Cell). The evidence on which the authors have made this conclusion is rather weak and possibly flawed. First of all, gene expressions on whole body for il34 distinguishing WT vs slbp1 mutant and that csf1r not being detected in a single cell analysis of microglial precursors cannot form a strong argument that this pathway is not involved as csf1r may just be lowly expressed and global expression may not be informative. The authors must analyze the il34 mutants to examine whether retinal microglia colonize or not in parallel with the tectal microglia. The only functional experiment they use is an rx1:il34 overexpression in the eye which was not sufficient to increase macrophage colonization-however the caveat is that this overexpression is mosaic and whether functional il34 proteins are getting produced is not known.

We appreciate this suggestion. A similar concern was raised by reviewer 2, in suggestion (2).

In accordance with reviewer 3’s suggestion, we investigated the number of ocular microglial precursors in zebrafish *il34* mutants. We found that the number of ocular microglial precursors was severely reduced in *il34* mutants. Very few microglial precursors (from 0 to 2 cells) are associated with blood vessels in *il34* mutant eyes at 48 hpf. These data suggest that IL34 is required for microglial precursor colonization of the retina in zebrafish. Since almost no microglial precursors can enter the optic cup, it is very likely that IL34 helps to guide microglial precursors from the yolk to the retina, upstream of blood vessel-mediated entry of the optic cup and neurogenesis-mediated infiltration of the neural retina. We have added these new data to Figure 6AB and have revised our conclusion: There are three steps for microglial colonization of the retina: In the first step, IL34 attracts microglial precursors and promotes their migration into the brain and eye. In the second step, blood vessels guide microglial precursors to enter the optic cup. In the third step, retinal neurogenesis enables microglial precursors to infiltrate the neural retina. This conclusive summary is also indicated in new Figure 6C.

7) Finally, more data points are required to formulate a reliable conclusion that microglia have a preference for differentiated retinal neurons and less with progenitors using the chimeric host-donor transplantation. There is a big differential in number of neural columns transplanted in mutants (more donor cells 3-7 columns in mutant hosts, than the 2-4 columns in control wt host group), which can misleadingly show an increased association of microglia with mature columns just because there are more columns in the slbp1 mutants. More Ns to establish equivalent range of donor transplantation (especially larger numbers in the wt group), and also the reciprocal test of putting donor mutant columns into WT host, will provide the needed comprehensive test of this concept using chimeras.

A similar concern was raised by reviewer 2, in suggestion (5).

We increased the number of wt->wt transplantation samples to 5. As reviewer 3 mentioned, it is ideal to maintain similar conditions across all samples regarding the total number of host microglial precursors per eye and the total number of transplanted donor columns per eye.

In the revised manuscript, we showed all this information plus the number of microglial precursors associated with transplanted retinal columns for each sample: 5 samples of wt-> wt and 3 samples of wt -> mut in Figure 5—figure supplement 1. The total number of host microglial precursors at 48 hpf was n=19.6 for wt-> wt and n=7.7 for wt-> mut, which is consistent with data shown in Figure 4AB. The total number of transplanted donor columns is n=5.6 for wt-> wt and n=4.7 for wt-> mut, which is not a big difference. Thus, we believe that our comparison of the fraction of microglial precursors associated with donor retinal columns between wt -> wt and wt -> mut was done appropriately to evaluate trapping efficiency for donor retinal cells by host microglial precursors.

8) The use of slbp1 mutants being only interpreted as a neurogenesis mutant can be problematic as this gene slbp1 (stem-loop binding protein binds to 3'end histone mRNAs) regulates degradation of histone proteins and replication-dependent synthesis and can have broad effects similar to the issue with hdac1 mutants. Complementary method to create neuron-specific disruption of neurogenesis would be ideal to substantiate the concept that the differentiation status of retinal neurons strongly affect colonization of microglia in the retina.

We appreciate this suggestion. Reviewer 2 raised a similar concern in suggestion (4).

We previously reported that Notch1 intracellular domain (NICD) suppresses retinal neurogenesis in zebrafish (Yamaguchi et al., 2005). To inhibit retinal neurogenesis more directly, without perturbation of microglial precursor functions, we overexpressed NICD in retinal cells and examined ocular microglial colonization of the retina. First, we confirmed that overexpression of NICD significantly suppresses retinal neurogenesis in zebrafish by injecting a DNA expression construct encoding UAS: myc-tagged NICD into *Tg[hs:gal4; ath5:EGFP]* double transgenic embryos (Figure 4—figure supplement 7).

Next, we established a zebrafish transgenic line, *Tg[rx1:gal4-VP16],* in which gal4-VP16 is expressed under the control of the *rx1* promoter, which drives mRNA expression in retinal progenitor cells. Then, we injected a mixture of two DNA expression constructs encoding UAS:myc-tagged NICD and UAS:EGFP into *Tg[rx1:gal4-VP16; mfap4:tdTMT-CAAX]* embryos. We selected embryos in which EGFP was expressed in most retinal cells at 24 hpf, and used them to investigate the number of ocular microglial precursors. We found that the number of ocular microglial precursors was significantly decreased in retinas overexpressing NICD, compared with retinas expressing only EGFP. Since the *rx1* promoter does not drive NICD expression in microglial precursors, these data suggest microglial colonization of the neural retina depends on retinal neurogenesis. We have added these new data to Figure 4EF.

In addition, we performed more sparse mosaic expression of NICD in wild-type retinas by injection of DNA construct encoding UAS:EGFP or a mixture of UAS:myc-NICD and UAS:EGFP into *Tg[hsp:gal4: mfap4:tdTMT-CAAX]* transgenic embryos. A fraction of microglial precursors associated with EGFP-positive columns was lower after UAS:NICD; UAS:EGFP injection than in UAS:EGFP injection only. Trapping efficiency of microglial precursors in each EGFP-positive columns was also lower in UAS:NICD; UAS:EGFP injection than in only UAS:EGFP injection. We have added these data to Figure 5EFGH.

Taken together, these data strongly support the possibility that retinal neurogenesis induces microglial precursors to infiltrate the retina.

[Editors’ note: what follows is the authors’ response to the second round of review.]

Overall, the authors have done a thorough and comprehensive job of addressing the points raised by the initial peer review and added additional controls and significant new data. This has particularly strengthened the conclusion that microglia colonization of retina is linked to neurogenesis, which is the central new finding of this study. However, there are a few remaining (minor) concerns that need to be addressed as summarized below by reviewers 1 and 3. In addition, there are few areas in the manuscript that overstate the conclusions based on the data reported in the manuscript. Please consider modifying the following conclusions it is not feasible to address experimentally in a timely manner.– The origin and characterization of the macrophage population(s) in the retina. For example, please state clearly their definition of microglia early in the results (In this manuscript, we define microglia as…").

We have added the following sentences on page 6, lines 130-133:

“In this study, we focus on two macrophage markers, mpeg1.1 and mfap4, and define mpeg1.1; mfap4positive cells inside the optic cup as microglial precursors colonizing the zebrafish retina.”

– The spatiotemporal role and relationship of IL34, microglia colonization and neurogenesis as raised by reviewer 1. These concerns can be addressed by adding experiments or modifying the current conclusions.

We have revised the text and Figure 6C in accordance with referee suggestions. Furthermore, we have examined the number of ocular microglial precursors in wild-type, *il34* heterozygous and homozygous mutants at an earlier stage (34 hpf) and confirmed that the number was zero and significantly lower in *il34* homozygous mutants than in wild-type and heterozygous mutants. We have added these data in new Figure 6—figure supplement 2. Accordingly, the original Figure 6—figure supplement 2 is renamed Figure 6figure supplement 3.

Reviewer #1:Please consider the following suggestions for the paper.– The authors should use other markers to verify that the cell populations of interest are microglial precursors. Using only macrophage markers may not encapsulate the heterogeneity of embryonic microglia in the zebrafish. Can the authors analyze other markers for microglia to demonstrate their identity? While they cite papers referencing apoeb, the data in the manuscript demonstrate microglia in different CNS regions have distinct properties. There are several reagents like antibodies against 4C4, or L-plastin or in situ hybridization of microglia genes like apoeb, or p2ry12 that could be used, as done in other zebrafish microglia manuscripts.

We agree that developing zebrafish larvae have a heterogenous population of microglia. Thus, we used a combination of macrophage/microglia markers *mpeg1.1* and *mfap4* throughout our study.

However, we understand the referee’s concern as to whether both *mpeg1.1* and *mfap4* marker-expressing cells represent microglial precursors, mature macroglia, or macrophages in the early stage of embryonic development in zebrafish. To clarify this point, we are currently investigating the state of mfap4:tdTomato-positive cells from 2 – 5 dpf, by scRNA-seq, using zebrafish heads at 2, 3 and 5 dpf. We found that 2 dpf, *mpeg1.1* and *mfap4*-expressing cells form only one cluster. This single cluster expresses both microglial and macrophage markers *ccl34b.1* and *lygl1*, but importantly does not express a mature microglial marker *apoeb.* Thus, it is very likely that 2 dpf *mpeg1.1* and *mfap4*expressing cells are macrophage/microglial precursors. At 3 dpf, *mfap4*-expressing cells are separated into 5 clusters. Among them, only two clusters express *apoeb*, suggesting that mature microglia appear by 3 dpf. Importantly, in these *apoeb*-positive clusters, the macrophage marker *lygl1* expression is downregulated and another microglial marker *ccl34b.1* expression is upregulated. On the other hand, *lygl1* expression is maintained and *ccl34b.1* expression is downregulated in other three *apoeb*-negative clusters. Thus, fate determination of microglia and macrophage proceeds at 3 dpf. At 5 dpf, cell fate for mature microglia and macrophages is more clearly segregated in *mfap4*-expressing cells. Thus, our current unpublished data revealed that *mpeg1.1*/*mfap4*-expressing cells at 2 dpf are reasonably classified as macrophage/microglial precursors, and a fraction of them differentiate into mature microglia after 3 dpf. Since there is a classic and pioneer work indicating that early macrophages outside the brain never express Apoe (Herbomel et al., 2001), we believe that *mpeg1.1*/*mfap4*-positive ocular cells at 2 dpf differentiate into mature microglia after 3 dpf. Thus, in this study, we focus on *mpeg1.1/mafap4* cells from 32 to 54 hpf, so we have defined *mpeg1.1/mafap4*-expressing cells in the optic cup as “microglia precursors” in our manuscript.

Consistent with our scRNA-seq data, labeling with anti-ApoEb antibody or in situ hybridization with *apoeb* RNA probe does not capture microglia precursors before 2 dpf in zebrafish. Furthermore, 4C4 antibody has not been shown to capture microglia precursors before 2dpf. p2ry12 expression also starts in *mpeg1.1/mfap4*-positive cells at 3dpf. Thus, none of them are available to detect microglial precursors for our study. However, in accordance with the referee’s suggestion, we examined the number of L-plastin-positive cells at 32 – 54 hpf by in situ hybridization, and confirmed that the number of L-plastin-positive cells is comparable with that of mpeg1.1:EGFP-positive cells shown by live imaging (Figure 1BC), although L-plastin is a pan-leukocyte marker and labels both differentiated neutrophils and macrophages (Meijer et al. (2008) *Dev. Comp. Immunol.* 32, 36–49.). We have added data regarding ocular L-plastin-positive cell temporal profile in new Figure 1—figure supplement 1.

Accordingly, the original Figure 1—figure supplement 1–4 is renamed “Figure 1—figure supplement 2–5.”

– The systemic inhibition of circulation is a concern because the manipulation is a global perturbation that impacts heart contraction. The data supporting the reliance of migrating microglial precursors on vasculature would be more convincing if the authors specifically inhibited the vasculature that they propose is utilized for microglia colonization. This concern is heightened by the observation that the animals in the tnnt2aMO look very unhealthy.

We applied a few chemical inhibitors such as SU5416, which was reported to inhibit blood vessel formation (Covassin et al., (2006), PNAS 103, 6554-9.; Herbert et al., (2009) Science 9, 294-298.); however, they affected myeloid cell genesis in developing zebrafish larvae, probably because both endothelial cells (blood vessel precursors) and myeloid cells are derived from hemangioblasts (Xiong (2008) *Dev Dyn* 237, 1218-1231.). So, there is no genetic or chemical tool for specific hyaloid blood ablation. We agree that *tnnt2a* morphants look unhealthy at 72 hpf, and we already noted in the legend of Figure 2—figure supplement 2AB that microglial colonization of the tectum may be different from previous reports, although the number of mfap4+ cells in the optic tectum is not significantly different between control morphants and *tnnt2a* morphants at 72 hpf, as reported previously. However, the embryonic condition of *tnnt2a* morphants is comparable to that of control morphants at 48 hpf (see Figure 2-figiure supplement 2C). Considering the live imaging observation that microglial precursors move into the optic cup along the hyaloid blood vessel, our conclusion that embryonic microglial precursors migrating into the retina depend on blood vessels is convincing.

– There are a few areas in the manuscript that overstate the conclusions based on the data reported in the manuscript. Please consider modifying the following conclusions in the manuscript:– The authors summarize their findings with a schematic in figure 6. The model states a stepwise process to microglia colonization of the retina, but the data reported in the manuscript support a more continuous process of microglia colonization. Further, the first step in the model is IL34-mediated migration to the brain, but the data reported in the manuscript only shows a reduction of microglia in the retina without follow up experiments to address when in the process microglia are disrupted in IL34 mutant animals. Please consider revising the model so that it is more consistent with the conclusions in the paper.– The comment "it is very likely that csf1r-il34 signaling promotes…" overstates the conclusion one can draw from the reported data. It would be more accurate to state that it is involved in colonization. Otherwise, the authors should assess the step wise process of colonization when IL34 is perturbed.

We agree with two of the referee’s concerns above. In accordance with his suggestions, we have revised Figure 6C and statements about the model in the Abstract, Discussion and Figure 6C legend, that IL34-mediated movement of microglial precursors into the brain may proceed in parallel or complement blood vessel and neurogenesis-mediated colonization of the retina, instead of stepwise regulation of the three processes.

– The authors utilize the NICD manipulation to test whether neurogenesis is required for microglia colonization of the retina. This is a clever approach to altering neurogenesis. With this experiment, it is possible that the Notch pathway is directly impacting microglia colonization. Please consider modifying the sentence line 379-380 "Thus microglial precursors are less attracted…in which neurogenesis was arrested." to add a disclaimer that the direct role of Notch signaling cannot be ruled out.

We introduced expression of NICD using UAS:NICD combined with *Tg[hsp:Gal4]* or *Tg[rx1:Gal4VP16]*. We observed decreased colonization of microglial precursors in retinas expressing NICD by *Tg[rx1:Gal4-VP16]* (Figure 4EF). Since *rx1* promoter drives Gal4-VP16 only in retinal progenitor cells, we can exclude the possibility that Notch signaling directly influences microglial precursors to affect colonization of the retina. In the experiment of Figure 5E-H, *hsp* promoter may drive NICD expression in microglia. However, in this experiment, we introduced UAS:EGFP together with UAS:NICD to monitor which cells express NICD. We have not observed EGFP expression of mfap4:tdTomato expressing microglial precursors in the images that we analyzed. Thus, we consider it very unlikely that Notch signaling directly influences microglial precursors to affect their colonization of the retina.

So, we would like to keep this conclusive sentence without addition of a disclaimer.

Reviewer #3:In this study, the authors investigate microglia colonization of the retina in zebrafish. Some of their observations confirm work from previous studies. They first carefully show that there is almost no proliferation of microglial precursors, which was previously shown in mouse retina (Santos et al., 2008) but not in zebrafish. They make the sound conclusion that microglia precursors migrate in to colonize the retina. They clearly show that neuronal apoptosis is not the major cue for microglial precursor colonization of the retina as it is for tectum, although this was previously documented in zebrafish by Wu et al., 2018 (which they gloss over in the manuscript). They also show that Csf1r-il34 signaling is important for microglia colonization of retina, although this was also shown by Wu et al. 2018 (again glossed over). Due to almost complete failure of microglia precursors to populate they eye in IL34 mutants, they argue that Csf1r-il34 signaling promotes microglial precursor movement from yolk to the optic cup upstream of the blood vessel-mediated guidance mechanism. However, it remains unclear which cell populations express IL34 developmentally to attract microglia precursors to the eye and at what stages this occurs. So their observations regarding the role of Csf1r-il34 signaling do not advance much beyond what is shown by Wu et al., 2018.They also make some new observations. They convincingly show that entry of microglial precursors into the optic cup through the choroid fissure depends on ocular blood vessels. Most interestingly, they show using three different manipulations that delaying retinal neurogenesis reduces microglial precursor colonization of the retina. They further show using elegant transplant studies that microglia precursors have greater affinity for differentiating neurons than for retinal progenitor cells. These experiments are carefully done and convincing. The mechanisms underlying this process are unclear since RGC differentiation begins at 27hpf, but microglia precursors don't enter until after 42hpf. They rule out IL34 as the signal since it is not altered in their slbp1 mutants (which show delayed delayed neurogenesis and delayed microglia colonization). So the main new conclusion is that microglia colonization of retina depends upon ocular blood vessels and is linked to neurogenesis. These findings are interesting and important.The authors have thoughtfully responded to prior comments from reviewers and added additional controls and significant new data. This has particularly strengthened the conclusion that microglia colonization of retina is linked to neurogenesis, which is the central new finding of this study. However, the mechanisms underlying this process remain undefined.

We appreciate these comments on our studies. Previously, Wu et al., (2018) demonstrated that microglial colonization of the brain, including the retina, depends on CSF-R and a CSF-R ligand, IL34. Following this pioneer work on zebrafish colonization of the brain, we have focused on the retina to investigate the microglial precursor colonization process in more detail. We found that microglial precursors enter the optic cup along the hyaloid blood vessel. Furthermore, surprisingly, microglial precursors infiltrate the neural retina through the neurogenic area, and a blockade of neurogenesis compromises microglial colonization of the retina. Although the mechanism underlying neurogenesis mediated colonization is unknown, we are planning to investigate it as our next project. As per the referee’s suggestion regarding Wu et al., (2018), we have cited their reference in the text.